# Learning Nonlinear Causal Reductions to Explain Reinforcement Learning Policies

**Armin Kekić** [1]    **Jan Schneider** [1]    **Dieter Büchler** [1,2,3,4]

**Bernhard Schölkopf**[*1,5,6]    **Michel Besserve**[*1,7]

[1] Max Planck Institute for Intelligent Systems, Tübingen, Germany
[2] University of Alberta, Canada
[3] Alberta Machine Intelligence Institute (Amii)
[4] CIFAR AI Chair
[5] Tübingen AI Center, Tübingen, Germany
[6] ELLIS Institute, Tübingen, Germany
[7] Technische Universität Braunschweig, Germany

`{armin.kekic,jan.schneider,dieter.buechler}@tue.mpg.de`
`{besserve,bs}@tue.mpg.de`

## ABSTRACT

Why do reinforcement learning (RL) policies fail or succeed? This is a challenging question due to the complex, high-dimensional nature of agent-environment interactions. We take a causal perspective on explaining the global behavior of RL policies by viewing the states, actions, and rewards as variables in a low-level causal model. We introduce random perturbations to policy actions during execution and observe their effects on the cumulative reward, learning a simplified high-level causal model that explains these relationships. To this end, we develop a nonlinear Causal Model Reduction framework that ensures approximate interventional consistency, i.e., the simplified high-level model responds to interventions in a way consistent with the original complex system. We prove that for a class of nonlinear causal models, there exists a unique solution that achieves exact interventional consistency, ensuring learned explanations reflect meaningful causal patterns. Experiments on both synthetic causal models and practical RL tasks - including pendulum control and robot table tennis - demonstrate that our approach can uncover important behavioral patterns, biases, and failure modes in trained RL policies.[1]

## 1 INTRODUCTION

In recent years, reinforcement learning (RL) has demonstrated remarkable successes in diverse domains, from achieving superhuman performance in games like Go [1] and Atari [2], to enabling sophisticated control in robotics [3]. The field has also seen significant adoption in real-world applications, including autonomous systems [4], recommendation engines [5], and process optimization [6]. As RL systems continue to be deployed in increasingly consequential settings, understanding the behavior and decision-making processes of trained policies becomes a practical necessity for ensuring reliability, safety, and trust. Hence, a natural and critical question arises: *"Why did a policy fail or succeed?"* These so-called *policy-level explanations* (PLE) [7] allow to assess the overall behavior and competency of an RL agent. Notably, as pointed out by Milani et al. [7, Section 8],

> *"When RL is applied to real-world domains, training directly in that domain may be prohibitively expensive or potentially unsafe. As a result, the agent is trained on a simulated domain, which is designed to be similar to the real-world domain but may differ in subtle ways. [...] In this case, it is critical that the human operator agrees with*

---

[*]Joint supervision.
[1]Code is available at: https://github.com/akekic/targeted-causal-reduction.git.

*the overall approach of the agent, [...] which requires understanding the entire policy. PLEs are best suited to this case [compared to other RL explainability approaches]."*

Trustworthy PLEs are challenging to obtain, as RL often uses parameter-rich neural networks to map from complex observation spaces to actions at each time step, with indirect consequences on the overall behavior that are not easily accessible to human intuition. Moreover, the credit assignment problem [8]—determining which actions contributed most significantly to eventual outcomes—is a fundamental obstacle to developing policy explanations. In particular, the dynamic dependencies between states and actions induce spurious correlations with the outcome, meaning that those features can be predictive of the outcome without causing it [9].

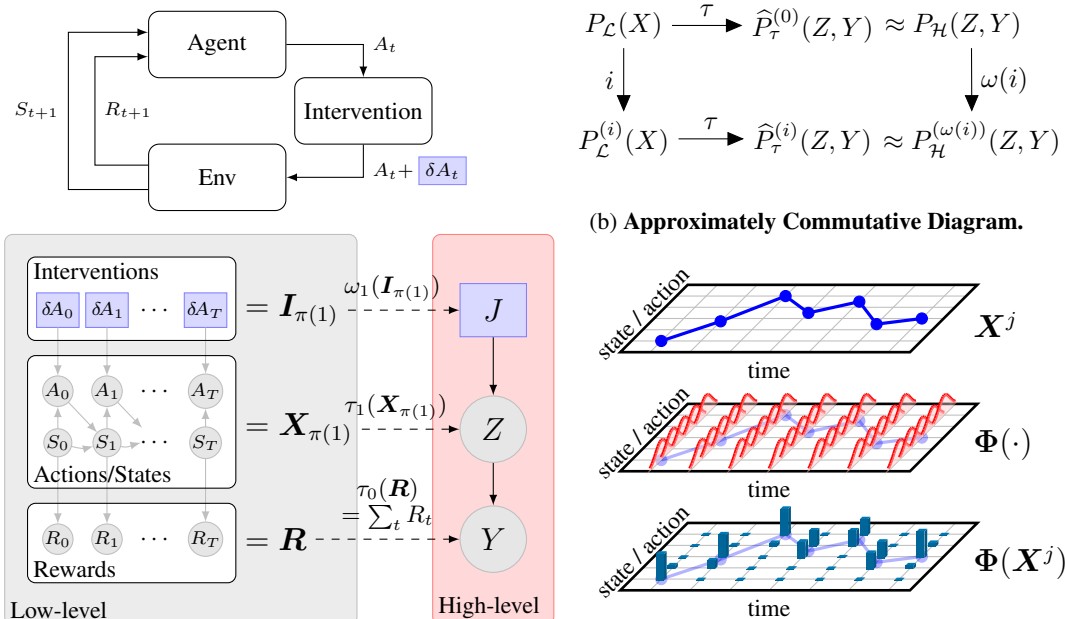

(a) **From RL Policies to Causal Explanations.**

(b) **Approximately Commutative Diagram.**

(c) **Interpretable Reduction Function Class.**

Figure 1: **Learning Causal Explanations of RL Policies.** (a) shows how RL policies are translated to a Causal Model Reduction problem. We sample episodes from the interactions between a trained agent and its environment, where the sampled actions are perturbed by random shifts $\delta A_t$. We treat the episode variables as nodes in a low-level causal graph, with $\delta A_t$ acting as shift interventions on the actions $A_t$. Map $\tau$ condenses the low-level causal variables to a simpler high-level model with two variables: the target $Y$, summarizing the rewards, and the cause $Z$, summarizing the subset $\pi(1)$ of low-level nodes. Similarly, $\omega_1$ maps the vector of low-level interventions $\boldsymbol{I}$ to its high-level counterpart $J$. (b) shows how the maps $\tau$ and $\omega$ are learned by making the diagram approximately commutative, minimizing the divergence between the distributions on each side of the $\approx$ sign. (c) shows a nonlinear interpretable function class that can be used to learn the reduction maps $\tau_1$ and $\omega_1$. A state/action variable's trajectory $\boldsymbol{X}^j$ is encoded through equally spaced Gaussian kernels to a feature vector $\boldsymbol{\Phi}(\boldsymbol{X}^j)$.

In this work, we take a causal perspective on PLE summarized in Fig. 1, focused on policies learned on simulations of real-world physical systems, with continuous state and action spaces. We formulate the problem as a *Causal Model Reduction* (CMR), where we treat the joint system of actions, environment variables, and rewards as a complex *low-level causal model*. To causally probe the low-level model, we introduce random perturbations to policy actions as interventions and observe their impact on cumulative rewards. We learn a mapping to a simplified *high-level causal model* with few "macroscopic" variables, summarizing the most important factors influencing the expected reward. As our main learning signal, we use *interventional consistency*: the low- and high-level models should respond to interventions in a similar fashion. The map from the original causal model to the reduced one can thus be used to explain the behaviors that are most influential on the success or failure of the RL policy.

Our contributions are: (i) formulating the problem of explaining RL policy behavior as a Causal Model Reduction and developing a nonlinear extension of Targeted Causal Reduction (TCR) [10] that learn high-level causes a target phenomenon in complex systems, (ii) providing theoretical guarantees of solution uniqueness for a broad class of nonlinear models, ensuring unambiguous explanations despite the identifiability challenges of nonlinear systems, (iii) introducing a class of interpretable nonlinear reduction functions that help explain the behavior captured by the learned reductions, (iv) demonstrating experimentally that our approach uncovers behavioral patterns and biases in two trained RL tasks.

## 2 BACKGROUND

**Notation.** We use lowercase letters for deterministic variables and capital letters for random variables. $X \sim P$ means $X$ has distribution $P$. We use boldface for column vectors, and $\boldsymbol{X}_S$ for the subvector of $\boldsymbol{X}$ restricted to the components in set $S$. The number of elements in a set $S$ is $\#S$.

### 2.1 STRUCTURAL CAUSAL MODELS (SCMS)

SCMs are a mathematical framework for representing cause-effect relationships in complex systems.

**Definition 2.1** (Structural Causal Model [11, 12]). An $n$-dimensional SCM is a triplet $\mathcal{M}=(\mathcal{G}, \mathbb{S}, P_{\boldsymbol{U}})$ consisting of: (i) a directed acyclic graph $\mathcal{G}$ with $n$ vertices, (ii) a joint distribution $P_{\boldsymbol{U}}$ over exogenous or noise variables $\{U_j\}_{j \leq n}$, (iii) a set $\mathbb{S}=\{X_j := f_j(\mathbf{Pa}_j, U_j), j=1, \ldots, n\}$ of structural equations, where $\mathbf{Pa}_j$ are the variables indexed by the set of parents of vertex $j$ in $\mathcal{G}$. This induces a joint distribution $P_{\mathcal{M}}$ over the endogenous variables $\boldsymbol{X} = [X_1, \ldots, X_n]^{\mathsf{T}}$.[2]

The endogenous variables $\boldsymbol{X}$ encode the system's observables, where each variable $X_j$ is determined through the deterministic causal mechanism $f_j$, its parent variables $\mathbf{Pa}_j$, and the exogenous noise variable $U_j$. One often encountered class of SCMs are *additive noise models* where structural equations take the simplified form $X_j := f_j(\mathbf{Pa}_j) + U_j$.

*Interventions* are encoded in SCMs by replacing one or several structural equations. An intervention transforms the original model $\mathcal{M}=(\mathcal{G}, \mathbb{S}, P_{\boldsymbol{U}})$ into an intervened model $\mathcal{M}^{(\boldsymbol{i})}=(\mathcal{G}^{(\boldsymbol{i})}, \mathbb{S}^{(\boldsymbol{i})}, P_{\boldsymbol{U}}^{(\boldsymbol{i})})$, where $\boldsymbol{i}$ is the vector parameterizing the intervention. The base probability distribution of the unintervened model is denoted $P_{\mathcal{M}}^{(0)}$ or simply $P_{\mathcal{M}}$ and the interventional distribution associated with $\mathcal{M}^{(\boldsymbol{i})}$ is denoted $P_{\mathcal{M}}^{(\boldsymbol{i})}$. In this work, we focus on *shift interventions*, which modify the structural equation of variable $X_l$ by shifting it by a scalar $i_l$

$$\{X_l := f_l(\mathbf{Pa}_l, U_l)\} \mapsto \{X_l := f_l(\mathbf{Pa}_l, U_l) + i_l\}. \tag{2.1}$$

These can be combined to form multi-node interventions with vector parameter $\boldsymbol{i}$. Shift interventions are particularly well suited to the case of simulators of physical systems with continuous state and action space, which is the main focus of the present work. In particular, they can be interpreted as an additional force or momentum in time-discretized equations of motion.

### 2.2 CAUSAL MODEL REDUCTIONS (CMRS)

Causal models with a large number of variables can be difficult to interpret and work with. Causal Model Reductions (CMRs) [10] are dimensionality reduction approaches that map such detailed low-level causal models to approximate high-level descriptions with fewer variables while preserving the essential causal properties. They closely relate to several notions of *causal abstraction* [13–19].

**Low- and High-level SCMs.** We consider two causal models:

- A *low-level SCM* $\mathcal{L}$ with endogenous variables $\boldsymbol{X} \sim P_{\mathcal{L}}$ (on range $\mathcal{X}$) with exogenous variables $\boldsymbol{U} \sim P_{\boldsymbol{U}}$, and set of possible interventions $\mathcal{I}$, leading to interventional distributions $(P_{\mathcal{L}}^{(\boldsymbol{i})})_{\boldsymbol{i} \in \mathcal{I}}$.
- A lower-dimensional *high-level SCM* $\mathcal{H}$ with endogenous variables $\boldsymbol{Z} \sim P_{\mathcal{H}}$ (on range $\mathcal{Z}$), exogenous variables $\boldsymbol{W} \sim P_{\boldsymbol{W}}$, set of interventions $\mathcal{J}$, and distributions $(P_{\mathcal{H}}^{(\boldsymbol{j})})_{\boldsymbol{j} \in \mathcal{J}}$.

---

[2]This SCM definition allows for confounding between variables through the potential lack of independence between the exogenous variables $\{U_j\}$. Although Def. 2.1 uses the common assumption of acyclic graphs for simplicity, our approach is also compatible with some families of causal graphs with cycles. See App. A.1.

$X$ is then *reduced* into a $\mathcal{Z}$-valued high-level random variable using a deterministic map $\tau : \mathcal{X} \to \mathcal{Z}$.

**Interventional Consistency.** The key criterion for a good reduction is *interventional consistency*: the high-level model should respond to interventions in ways that correspond to the original model's behavior under analogous interventions (*cf.* [13, 18–20]). To formalize this, we consider $\tau_{\#}[P_{\mathcal{L}}]$, the so-called *push-forward distribution* by $\tau$ of the low-level model distribution, such that

$$\tau(\boldsymbol{X}) \sim \tau_{\#}[P_{\mathcal{L}}] . \tag{2.2}$$

Interventional distributions are pushed forward in the same way from the low-level to the high-level, allowing to define the notion of an *exact transformation*:

**Definition 2.2** (Exact transformation [19]). A map $\tau : \mathcal{X} \to \mathcal{Z}$ is an exact transformation from $\mathcal{L}$ to $\mathcal{H}$ if it is surjective, and there exists a surjective *intervention map* $\omega : \mathcal{I} \to \mathcal{J}$ such that for all $\boldsymbol{i} \in \mathcal{I}$

$$\tau_{\#} \left[ P_{\mathcal{L}}^{(\boldsymbol{i})} \right] = P_{\mathcal{H}}^{(\omega(\boldsymbol{i}))} . \tag{2.3}$$

That is, regardless of whether we first intervene through $\boldsymbol{i}$ in the low-level model $\mathcal{L}$ and then map to the high-level $\mathcal{H}$, or first map to $\mathcal{H}$ and intervene with $\omega(\boldsymbol{i})$, we arrive at the same distribution.

## 2.3 Targeted Causal Reduction (TCR)

While exact transformations are theoretically elegant, they present two practical challenges: (i) achieving perfect equality in Eq. (2.3) is often infeasible, so we need a systematic method to learn approximately consistent reductions from data, and (ii) there are many possible consistent reductions and finding a meaningful one needs additional constraints. *Targeted Causal Reduction* (TCR) [10] addresses these challenges by focusing on explaining a specific target variable of interest and providing a concrete learning objective. Below, we describe the case of a single high-level cause, the general case is described in App. A.2.

**Definition 2.3** (General TCR Framework). In TCR, we learn transformations $(\tau, \omega)$ between models $\mathcal{L}$ and $\mathcal{H}$ under the following assumptions:

1. **Target**: $\mathcal{H}$ contains only two nodes: a predefined scalar target variable $Y = \tau_0(\boldsymbol{X})$ that quantifies a phenomenon of interest, and a high-level cause $Z = \tau_1(\boldsymbol{X})$ that needs to be learned.
2. **Parameterized High-level Models**: The high-level SCM is constrained to a class of linear additive Gaussian noise models $\{\mathcal{H}_{\boldsymbol{\gamma}}\}_{\gamma \in \Gamma}$ with parameters $\boldsymbol{\gamma}$ to be learned, implying that (i) the mechanism $Z \to Y$ is linear, with (linear) causal coefficient $\alpha$ and additive noise $W_0$ and (ii) the exogenous variables $\{W_0, W_1 = Z\}$ have a factorized Gaussian distribution $P_{\boldsymbol{W}} = P_{W_0} P_{W_1}$.
3. **Constructive Transformations**: The respective components of the maps $\tau$ and $\omega$ associated to the high-level cause and effect depend on disjoint subsets $\pi(1)$ and $\pi(0)$ of low-level variables:

$$\tau(\boldsymbol{x}) = (\bar{\tau}_0(\boldsymbol{x}_{\pi(0)}), \bar{\tau}_1(\boldsymbol{x}_{\pi(1)})) , \qquad \omega(\boldsymbol{i}) = (0, \bar{\omega}_1(\boldsymbol{i}_{\pi(1)})) . \tag{2.4}$$

the first component of $\omega$ is set to zero to prevent direct high-level interventions on $Y$, forcing changes in $Y$ to be explained through high-level interventions on $Z$.
4. **Approximate Consistency Objective:** Rather than requiring exactness of Eq. (2.3), TCR minimizes the consistency loss, quantifying an average discrepancy D between the push-forward interventional distributions and their corresponding high-level model distributions:

$$\mathcal{L}_{\text{cons}}(\tau, \omega, \gamma) = \mathbb{E}_{\boldsymbol{i} \sim P_{\boldsymbol{I}}} \left[ \text{D} \left( \widehat{P}_{\tau}^{(\boldsymbol{i})}(Y, Z) \| P_{\mathcal{H}}^{(\omega(\boldsymbol{i}))}(Y, Z) \right) \right] , \text{ with } \widehat{P}_{\tau}^{(\boldsymbol{i})} = \tau_{\#} \left[ P_{\mathcal{L}}^{(\boldsymbol{i})} \right] . \tag{2.5}$$

The relationship between those distributions and the learned reduction maps is shown in Fig. 1b. The expectation is taken over a prior distribution of interventions $P_{\boldsymbol{I}}$. [3]

---

[3]Remarks: In practice, $\pi(0)$ and $\pi(1)$ can be fixed based on domain knowledge, with $\pi(1)$ typically encompassing all variables potentially influencing the target that are not in $\pi(0)$. The intervention prior $P_{\boldsymbol{I}}$ can be uninformative (*e.g.* i.i.d. Gaussian) or informed by domain expertise.

**Intuition behind TCR.** TCR provides a form of *etiological explanation* for the target phenomenon $Y$, identifying the origins of its variation. It accomplishes this by creating an interpretable formulation where changes in $Y$ due to low-level interventions are explained through a high-level causal mechanism. TCR forces explanations to flow through the learned high-level cause $Z$. When asked *"What causes changes in $Y$?"*, TCR answers with a specific causal statement: *"$Y$ is changed by acting on the high-level cause $Z = \tau_1(X_{\pi(1)})$ through intervention $\omega(i_{\pi(1)})$"*. The interpretability of the maps $\tau$ and $\omega$ provides valuable insights: $\tau$ reveals which properties of the system influence $Y$, while $\omega$ shows which types of interventions most effectively change these properties.

**Linear TCR.** The TCR formulation by Kekić et al. [10] imposes some significant constraints: (i) **Linear transformations** for $\tau$ and $\omega$, simplifying analysis and facilitating interpretability but limiting expressivity; and (ii) **Gaussian KL-divergence approximation** for the discrepancy D in Eq. (2.5), where distributions are approximated as Gaussians with matched means and variances (see App. A.3). This has the benefits of taking a differentiable closed-form and allowing easy optimization.

## 3 FROM RL POLICIES TO CAUSAL MODEL REDUCTION

RL episodes inherently form causal chains where the agent's observations lead to actions, which cause environmental changes resulting in new states and rewards. These interactions naturally map to structural causal models. To identify the causal impact of specific actions on overall performance, we need to observe counterfactual outcomes—what would have happened if the agent had acted differently. We achieve this by introducing controlled perturbations to the policy's chosen actions and observing their effects. Specifically, we construct episodes where each action $A_t$ selected by the policy is perturbed by a small random shift $\delta A_t \sim \mathcal{N}(0, \sigma_t)$ before being executed in the environment:

$$(S_0, \, A_0 + \delta A_0, \, R_1, \, S_1, \, A_1 + \delta A_1, \, R_2, \, \ldots, \, S_{T-1}, \, A_{T-1} + \delta A_{T-1}, \, R_T, \, S_T) \, . \tag{3.1}$$

These perturbations serve as shift interventions in our causal framework and allow us to observe how deviations from the policy's intended actions propagate to the cumulative reward.

To formalize this as a TCR problem, we define the low-level endogenous variables $X$ consisting of states and actions $X_{\pi(1)} = (A_0, \ldots, A_{T-1}, S_0, \ldots, S_T)^\intercal$ and rewards $X_{\pi(0)} = R = (R_1, \ldots, R_T)^\intercal$, as shown in Fig. 1a. The interventions are $I_{\pi(1)} = (\delta A_0, \ldots, \delta A_{T-1}, 0, \ldots, 0)^\intercal$, where we intervene only on actions and not on states, and the target variable is the cumulative reward $Y = \sum_{t=1}^{T} R_t$.

With this formulation, we can now apply the TCR framework to learn a simplified high-level model with one high-level cause $Z$ that best explains variations in the target variable $Y$. The reduction maps $\tau_1$ and $\omega_1$ identify which aspects of the states and actions most significantly influence the overall performance, thereby providing an interpretable causal explanation of policy behavior.

## 4 NONLINEAR TARGETED CAUSAL REDUCTION (nTCR)

While the linear TCR framework introduced in Sec. 2.3 provides a principled approach to learning simplified causal models, many real-world phenomena - particularly in reinforcement learning - involve inherently nonlinear relationships that cannot be adequately captured by linear maps. In this section, we extend TCR to nonlinear settings while preserving its key benefits of interpretability and causal consistency. This extension, which we call nonlinear TCR (nTCR), allows us to discover and represent causal patterns in complex interactive systems such as those controlled by RL policies. Despite the added flexibility of nonlinear functions, we show that our approach maintains strong theoretical guarantees. Specifically, we demonstrate both the existence and uniqueness of reductions with exact interventional consistency for a broad class of nonlinear causal models.

### 4.1 NORMALITY REGULARIZATION

The original discrepancy of linear TCR in Eq. (2.5) does not enforce distributions to fully match (see App. A.3), and the nonlinear reduction can therefore lead to a highly non-Gaussian high-level cause. We therefore introduce a normality regularization encouraging $\tau_1(X)$ to be Gaussian. In addition to being more faithful to the initial idea of Def. 2.2, this allows for easier interpretation of

the learned causes (as it enforces a simple unimodal cause distribution). The theory behind the choice of consistency objective is further discussed in App. D.

Given the push-forward distribution $\widehat{P}_\tau^{(i)}(Z)$ (with CDF $F_{\widehat{P}_\tau^{(i)}(Z)}(x)$) of the high-level cause under intervention $i$, we first standardize this distribution to zero mean and unit variance to obtain $\widehat{P}_{\tau,\text{std}}^{(i)}(Z)$. We then measure the deviation from normality as the 1-Wasserstein distance $W_1$ between this standardized distribution and the standard normal distribution $\mathcal{N}(0,1)$ (with CDF $\Phi(x)$):

$$\mathcal{L}_{\text{norm}} = \mathbb{E}_{i \sim P(i)}\left[W_1(\widehat{P}_{\tau,\text{std}}^{(i)}(Z), \mathcal{N}(0,1))\right] = \mathbb{E}_{i \sim P(i)}\left[\int_\mathbb{R} |F_{\widehat{P}_{\tau,\text{std}}^{(i)}(Z)}(x) - \Phi(x)| \, \mathrm{d}x\right]. \quad (4.1)$$

The total optimization objective for nTCR then becomes $\mathcal{L}_{\text{total}} = \mathcal{L}_{\text{cons}} + \eta_{\text{norm}}\mathcal{L}_{\text{norm}}$. Here, $\eta_{\text{norm}}$ is a hyperparameter that controls the strength of the normality regularization, balancing the trade-off between achieving interventional consistency and maintaining Gaussian-like distributions for the high-level cause. In practice, we compute $\mathcal{L}_{\text{norm}}$ using a differentiable approximation of the CDF and by evaluating the $L_1$ distance at a finite set of points, as presented in App. C.1. This approach allows for efficient optimization while maintaining the desired distributional properties of the high-level causes.

## 4.2 EXACT SOLUTIONS

Nonlinear TCR allows a much larger space of functions to fit the learning objective. This can lead to overfitting and even potential non-identifiability issues where multiple reductions could exist, which would strongly limit the interpretability of the approach. The following statement provides identifiability guarantees even in the nonlinear case, as long as we obtain exact solutions.

**Proposition 4.1** (Uniqueness). *Assume (i) the low-level model is an additive noise SCM of the form*

$$\begin{bmatrix} \boldsymbol{X}_{\pi(1)}, \\ \boldsymbol{X}_{\pi(0)} \end{bmatrix} := \begin{bmatrix} \boldsymbol{f}_1(\boldsymbol{X}_{\pi(1)}) + \boldsymbol{U}_{\pi(1)} + \boldsymbol{i}_{\pi(1)}, \\ \boldsymbol{f}_0(\boldsymbol{X}_{\pi(0)}, \boldsymbol{X}_{\pi(1)}) + \boldsymbol{U}_{\pi(0)} \end{bmatrix}, \quad \boldsymbol{U}_{\pi(1)} \sim P_1 \text{ indep. of } \boldsymbol{U}_{\pi(0)} \sim P_0, \quad (4.2)$$

*(ii) $P_1$ admits a probability density with non-vanishing Fourier transform, (iii) The high-level model has a non-zero causal effect ($\alpha \neq 0$). Then, if there exists a constructive transformation following Def. 2.3 that is exact for all $\boldsymbol{i}_{\pi(1)} \in \mathbb{R}^{\#\pi(1)}$, it is also unique up to multiplicative and additive constants.*

See App. B.1 for the proof. The above result does not guarantee the existence of such exact transformations. The following proposition shows we can construct families of low-level additive noise models that admit a nonlinear exact transformation $\tau_1(\boldsymbol{X}_{\pi(1)}) \to \tau_0(\boldsymbol{X}_{\pi(0)})$.

**Proposition 4.2** (Existence). *Assume a low-level additive SCM following Eq. (4.2). Assume additionally that (i) $U$ is jointly Gaussian, (ii) $\boldsymbol{f}_0(\boldsymbol{X}_{\pi(0)}, \boldsymbol{X}_{\pi(1)}) = \boldsymbol{h}_0(\boldsymbol{X}_{\pi(0)}) + B(\boldsymbol{X}_{\pi(1)} - \boldsymbol{f}_1(\boldsymbol{X}_{\pi(1)}))$, for some functions $\boldsymbol{h}_0, \boldsymbol{f}_1$, some matrix $B \in \mathbb{R}^{(\#\pi(0) \times \#\pi(1))}$, and (iii) $Y = \tau_0(\boldsymbol{X}_{\pi(0)}) = \boldsymbol{a}^\top(\boldsymbol{X}_{\pi(0)} - \boldsymbol{h}_0(\boldsymbol{X}_{\pi(0)}))$ for arbitrary non-vanishing $\boldsymbol{a} \in \mathbb{R}^{\#\pi(0)}$, then the transformation defined by the following maps is exact*

$$\bar{\tau}_1(\boldsymbol{X}_{\pi(1)}) = \boldsymbol{a}^\top B(\boldsymbol{X}_{\pi(1)} - \boldsymbol{f}_1(\boldsymbol{X}_{\pi(1)})), \quad \bar{\omega}_1(\boldsymbol{i}_{\pi(1)}) = \boldsymbol{a}^\top B \boldsymbol{i}_{\pi(1)}. \quad (4.3)$$

See App. B.2 for the proof. We do not expect to cover all possible nonlinear exact transformations with this class of models. Neither do we expect that simulations of real-world systems considered in applications satisfy the above conditions and therefore admit an exact transformation.[4] However, the above results can be used to validate our algorithms, as discussed in Sec. 5.1.

## 4.3 INTERPRETABLE NONLINEAR FUNCTION CLASS

When the reduction is allowed to be nonlinear, it can be challenging to interpret what the high-level cause and intervention represent in the system without constraints on the mappings $\tau_1$ and $\omega_1$. For example, RL episodes possess inherent temporal structure that we can exploit to build more interpretable function classes for these mappings. We can express $\mathcal{X}$ as a Cartesian product across

---

[4]We thus use an approximate consistency objective to find explanation that are "as consistent as possible".

$d$ features, with each feature represented as a time series $\mathcal{X} = \mathcal{X}^1 \times \cdots \times \mathcal{X}^d$, where $\mathcal{X}^j$ represents the space of possible trajectories for the $j$-th feature over time. Each feature's trajectory space can be further decomposed as $\mathcal{X}^j = \mathcal{X}_1^j \times \cdots \times \mathcal{X}_T^j$. The intervention space $\mathcal{I}$ can be decomposed similarly. Leveraging this structure, we define our interpretable nonlinear function class for $\tau_1$ as:

$$\tau_1(\boldsymbol{X}) = \sum_{j=1}^{d} \tau_1^j(\boldsymbol{X}^j) = \sum_{j=1}^{d} \sum_{t=1}^{T} w_{j,t} \cdot \phi_{j,t}(X_t^j) = \sum_{j=1}^{d} \sum_{t=1}^{T} w_{j,t} \cdot \exp\left(-(x - \mu_{j,t})^2 / 2\sigma_{j,t}^2\right) \quad (4.4)$$

Where $w_{j,t}$ are learned weights, and $X_t^j$ is the value of feature $j$ at time $t$. For our implementation, we use Gaussian kernels $\phi_{j,t}$ as the basis functions, where $\{\mu_{j,t}, \sigma_{j,t}\}$ are fixed parameters, set to span the range of values typically encountered, see Fig. 1c. The kernel width $\sigma_{j,t}$ and the number of kernels control the bias-variance trade-off: wider kernels and fewer kernels both increase smoothing (higher bias, lower variance).[5] This can be viewed as a continuous one-hot encoding of variable-value pairs across the episode, allowing for smooth interpolation between similar states. This approach ensures that our nonlinear reduction remains interpretable, as we can identify which features, at which time steps, contribute most significantly to the high-level causal explanation by examining the learned weights $w_{j,t}$.

## 5 CASE STUDIES

### 5.1 EXPERIMENTS ON SYNTHETIC DATA

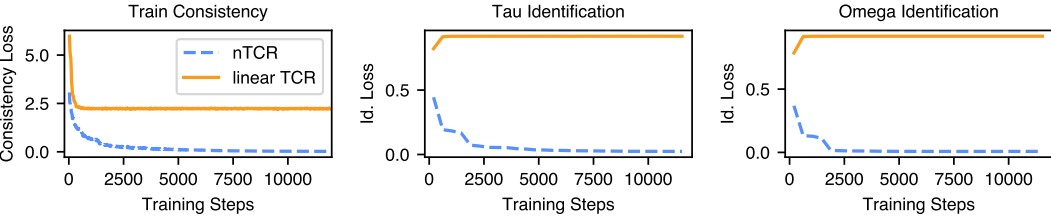

Figure 2: **Identification of Ground-Truth Solutions for Synthetic Low-level Models.** Consistency loss (left) and the identification losses measuring agreement with the ground-truth solutions (definition in App. E.1) for the $\tau$- and $\omega$-functions (middle and right) over the reduction training run.

Before applying our approach to RL problems, we first validate its theoretical properties on synthetic data generated from known low-level causal models. This controlled setting serves two important purposes: (i) it demonstrates that for this model class, nTCR can indeed find the unique solution that perfectly minimizes the causal consistency loss, confirming our theoretical guarantees; and (ii) it validates that our training methodology successfully converges to the correct solution.

We generate synthetic data from 10 randomly sampled low-level causal models following the structure described in Prop. 4.2 with $\dim(\boldsymbol{X}_0) = 2$ and $\dim(\boldsymbol{X}_1) = 9$. Each entry in $\boldsymbol{a}$ and $B$ are sampled from a standard normal distribution. For these experiments, we use neural networks as generic function approximators for both $\tau_1$ and $\omega_1$. For more details on the experimental setting, please refer to App. E.1. Fig. 2 shows the consistency loss and an identification measure that quantifies how closely the learned $\tau$- and $\omega$-maps match the ground truth solutions.[6] We observe that nTCR successfully finds reductions with near-zero consistency loss and converges to the theoretical solution outlined in Prop. 4.2, validating both our theory and implementation.

### 5.2 PENDULUM

Pendulum [21] is a classic swing-up control task where a pendulum must be brought to the upright position. With limited torque, the agent must develop momentum through strategic oscillations rather than directly moving to the goal position. The state space comprises three variables: x-y coordinates $(\cos(\theta), \sin(\theta))$ and angular velocity, while the action is the applied torque. The reward

---

[5]The function $\omega_1$ is defined analogously, operating on the intervention space $\mathcal{I}_{\pi(1)}$.

[6]A normalized L2 loss between the learned and ground-truth high-level cause; see App. E.1 for more details.

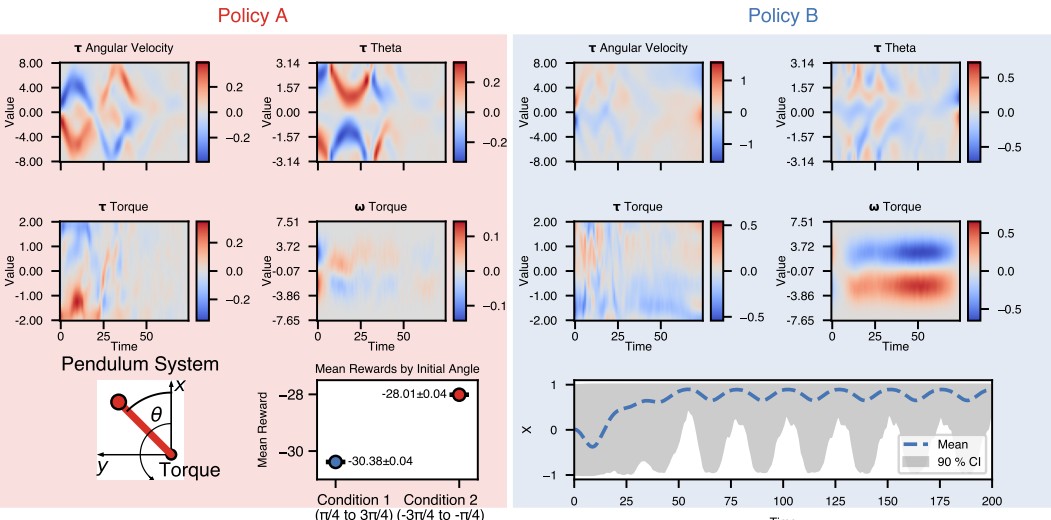

Figure 3: **Pendulum Task.** The top two rows show the learned nTCR $\tau$- and $\omega$-maps for two policies A and B. The heatmaps show the learned reductions $\tau_1^j(\boldsymbol{x}^j)$ and $\omega_1^j(\boldsymbol{i}^j)$, where $j$ indexes the state/action variables (angular velocity, theta, and torque). Since we only intervene on torque, it is the only variable with a nonzero $\omega$-map. The bottom left plot shows the pendulum system setting. The middle plot on the bottom row shows the mean reward for Policy A for pendulums starting in the left quadrant (Condition 1) and those starting in the right quadrant (Condition 2). The standard error of the mean is shown (error bars are smaller than the data point in the plot). The bottom right plot shows the mean $x$-position of the pendulum for episodes under Policy B and the $90\%$ confidence interval.

function penalizes deviation from the upright position, angular velocity, and action magnitude. For the reductions, we choose the more natural feature $\theta$ instead of the $x$- and $y$-positions. More details on the experiment setting, and the agent and nTCR training are given in App. E.2.[7]

Fig. 3 (left, Policy A) shows that nTCR identifies two trajectory groups with significant reward variations: pendulums starting in the bottom-right corner swinging clockwise (red, higher rewards) versus those starting bottom-left swinging counterclockwise (blue, lower rewards). The reduction shows that the agent performs better clockwise than counterclockwise  -  a surprising bias given that initial conditions are uniformly sampled and the environment has mirror symmetry along the axis of gravitation.

For Policy B (Fig. 3, right), the learned omega map reveals that shifting torque to more negative values would improve performance, particularly towards the end of the episode. Analysis confirms this (Fig. 3, bottom-right): the policy fails to consistently stabilize the pendulum upright, allowing it to tip over in the positive $\theta$ direction before applying positive torque for a full rotation recovery. nTCR correctly identifies that applying more negative torque at the top position would prevent this instability.

## 5.3 TABLE TENNIS

We evaluate our approach on a robot table tennis simulation based on the real-world setup developed in [22, 23]. The environment features a four degrees-of-freedom robot arm actuated by pneumatic artificial muscles, positioned on one side of a table tennis table, shown in Fig. 4 (a,b). The robot's task is to return incoming balls launched by a ball gun to a desired landing point on the opponent's side of the table. The state space includes the robot state (joint angles, joint velocities, and air pressures in each muscle) and the ball state (position and velocity). The action space consists of the changes in desired pressures for each of the eight pneumatic muscles. The task requires precise timing and positioning to successfully intercept the ball and direct it toward the target. More details on the environment, the RL- and nTCR-training are given in App. E.3.[8]

---

[7]Example episodes for the two policies are shown in the supplementary video files.

[8]The behavior of the trained policy can be seen in the supplementary video files.

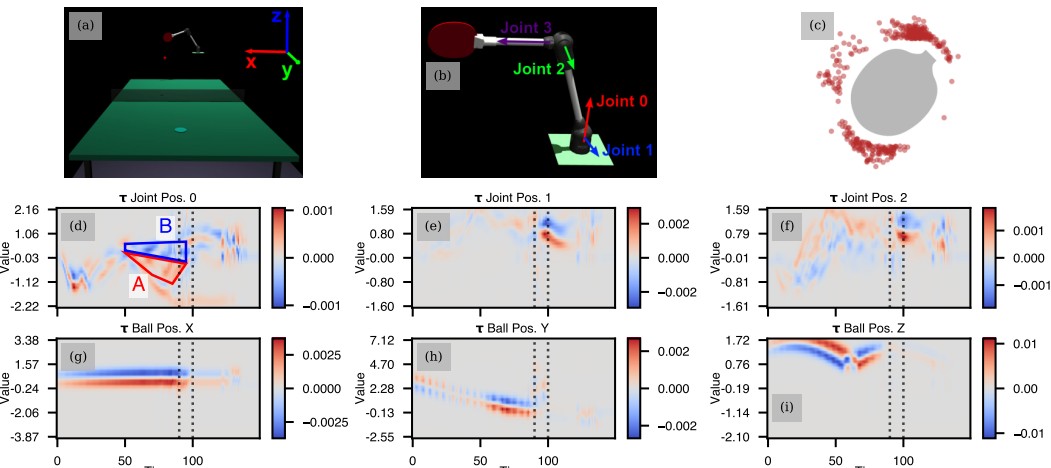

Figure 4: **Table Tennis Task.** A robot arm is trained to return incoming balls to a target location on the opponent's side of the table (a). The robot arm and its rotational axes are shown in (b). (c) shows the positions of 400 balls that the robot missed relative to the racket. (d-i) show the learned $\tau$ reduction maps for the joint positions 0-2 and the ball position. The reduction maps for the other variables are shown in App. E.3. The dotted vertical lines show the time range where the robot typically hits the ball.

The learned $\tau$ maps for the joint positions 0-2 and the ball positions are shown in Fig. 4. We make two key observations: one regarding the movements of the robot arm and another with respect to performance differences across ball trajectories. Joint 0 is the main axis used to swing the robot arm to accelerate the ball (see Fig. 4 (b)). The $\tau$-map for this variable is shown in Fig. 4 (d). The map highlights two critical state regions for joint position 0 during the period when the arm is swinging towards the ball (regions A and B in Fig. 4 (d)): (A) the arm swings back (negative angle values) before the ball arrives and forward when the ball arrives, leading to high rewards, and (B) the arm swings forward too early, leading to low rewards.

The TCR reductions for the ball position reveal two clear trends: (i) balls that travel further toward the outside edge of the table are more difficult to hit (Fig. 4 (g)), and (ii) balls that bounce further from the robot, or closer to the net, present greater challenges (Fig. 4 (h, i)). We validate these findings with an analysis of the missed balls, shown in Fig. 4(c). We observe that more balls miss the racket toward the top compared to the bottom, and more to the left (corresponding to the direction toward the outside of the table) than to the right. These observations are consistent with the $\tau$ reduction values for joint positions 1 and 2, shown in Fig. 4 (h, i). Around the time of ball contact, they have their largest positive contributions near $\pi/4$—corresponding to a relatively high and further out racket—and their largest negative contributions for larger angles—corresponding to a lower racket closer to its base.

## 6 RELATED WORK

Explainable reinforcement learning (XRL) focuses on understanding the decision-making processes of learning RL agents. Milani et al. [7] propose a taxonomy for XRL approaches into three categories: *feature importance* methods explaining the immediate context for single actions (*e.g.* [24–26]), *learning process* methods that reveal influential experiences from training (*e.g.* [27–29]), and PLEs [30–32], which is the category our approach belongs to. Specifically, nTCR belongs to the subcategory of techniques that extract abstract states, as we reduce the dimensionality of the states, actions, and reward spaces into high-level variables. A notable causal approach to XRL is Madumal et al. [33], who generates contrastive explanations for agent behavior through counterfactual analysis, though their approach differs from ours in that they require a predefined causal graph structure and focus on explaining individual actions rather than identifying high-level causal patterns across episodes. Also related to nTCR is [34] who rely on perturbations of the agent's policy to identify the time steps where the agent's control is crucial. Our approach goes further to extract a representation of what the policy actually does, with a description of the states and actions that benefit or hinder the global performance.

Our approach is an extension of linear TCR [10] which is itself grounded in theoretical work on causal abstraction [13–16, 19]. While these works establish formal conditions for valid causal abstractions, few have addressed the challenge of learning such abstractions from low-level systems. Geiger et al. [17, 35] focus on language models, where high-level variables and interpretations are already available and used to constrain a neural network's structure. Our approach, in line with Kekić et al. [10], takes the opposite direction: high-level abstractions are built from the ground up. This is conceptually related to causal feature learning [18] but differs in two key aspects: the use of shift interventions, and building continuous rather than strictly discrete high-level representations. Linear TCR was previously applied by Kekić et al. [10] only to scientific simulations. The similarities between simulating physical systems and RL environments motivated us to study and adapt this framework to the context of policy explanation.

Another related field is the study of Markov Decision Process (MDP), which model the systems we want to control with RL-agents. Several works [36–38] have taken a causal perspective to study MDPs and related models in order to analyze policy optimization, and notably the relevance of specific state variables for those policies. These methods can be leveraged to learn causal representations that are suitable for training RL-agents, which contrasts with our goal of interpreting already trained policies.

## 7 Discussion

We have introduced nTCR, a nonlinear dimensionality reduction framework for causal models, and used it to generate PLEs for trained policies. We have experimentally demonstrated that nTCR (1) can learn causally consistent explanations when they exist (as predicted by our theoretical results), (2) can identify biases and failure modes in RL policies on Pendulum and a robot table tennis simulation.

Our framework is subject to a number of constraints: (1) it makes linear and Gaussian high-level model assumptions that may need to be dropped in some applications, although it poses further interpretability challenges; (2) it requires to sample many perturbed episodes of interactions with the RL agent, which makes it mainly suited to scenarios where the agent is trained in simulation before real-world transfer; (3) it uses shift interventions, suited to continuous action spaces of the real-world applications we focus on, and adaptation to discrete action spaces could be considered, *e.g.* applying shift interventions to the logits of the network implementing the categorical policy used to sample discrete actions.

While nTCR overcomes the limitations of linear TCR by allowing for arbitrary functions as reduction maps, one of the central challenges for methods generating explanations is balancing expressivity with interpretability. Our Gaussian kernel-based function class (Sec. 4.3) represents a middle ground, offering more expressivity than linear maps while maintaining interpretability. More complex or less structured state/action data, such as for policies leveraging image inputs, may require additional steps for representing them appropriately, *e.g.* segmentation methods [39] or saliency maps [24, 40].

The applicability of our method primarily to simulated environments makes sense because it is designed for scenarios where policies can be safely probed through interventions, primarily in simulators before real-world deployment (*e.g.* in robotics or autonomous systems). While we focus on simulation due to safety and cost, the framework could, in principle, be applied to real systems where controlled interventions are feasible, and frameworks that combine simulated and real data [41].

Barring these limitations, nTCR provides model-agnostic causal PLEs that can notably be used to assess the behavior of RL agents in simulations before their potential transfer to the real world.

### Acknowledgments

Jan Schneider is supported by the Konrad Zuse School of Excellence in Learning and Intelligent Systems (ELIZA), sponsored by the Federal Ministry of Education and Research.

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

# Appendix

## CONTENTS

## A  ADDITIONAL BACKGROUND

### A.1  CYCLIC STRUCTURAL CAUSAL MODELS

**Definition A.1** (SCM (with cycles allowed)). An $n$-dimensional SCM is a triplet $\mathcal{M} = (\mathcal{G}, \mathbb{S}, P_{\boldsymbol{U}})$ consisting of:

- a joint distribution $P_{\boldsymbol{U}}$ over exogenous variables $\{U_j\}_{j \leq n}$,
- a directed graph $\mathcal{G}$ with $n$ vertices,
- a set $\mathbb{S} = \{X_j := f_j(\mathbf{Pa}_j, U_j), j = 1, \dots, n\}$ of structural equations, where $\mathbf{Pa}_j$ are the variables indexed by the set of parents of vertex $j$ in $\mathcal{G}$,

such that for almost every $u$, the system $\{x_j := f_j(\mathbf{pa}_j, u_j)\}$ has a unique solution $x = g(u)$, with $g$ measurable.

The unique solvability condition allows to consider a very general class of SCMs by allowing cycles ($\mathcal{G}$ may not be a DAG, while a DAG would imply unique solvability). Moreover, we allow hidden confounding through the potential lack of independence between the exogenous variables $\{U_j\}$. See Bongers et al. [42] for a thorough study of these models. Under these conditions, the distribution $P_U$ entails a well-defined joint distribution over the endogenous variables $P(X)$. By unique solvability, we define the mapping from exogenous to endogenous variables of this model under interventions as $M^{(i)} : u \mapsto x$ such that $M^{(i)}(U) \sim P_{\mathcal{M}}^{(i)}(X)$.

## A.2 Targeted Causal Reduction (Multiple Causes)

In this paper, we focus on a single high-level cause for Targeted Causal Reduction, as this is case used to develop our RL explainability method. Here, we present the more general TCR formulation with multiple high-level causes.

**TCR Framework with Multiple Causes.** The general TCR framework with multiple causes extends the single-cause version as follows:

1. **Target-oriented Structure**: We designate a scalar target variable $Y = \tau_0(X)$ that quantifies a phenomenon of interest. The high-level model has $(n+1)$ endogenous variables $\{Y, Z_1, \ldots, Z_n\}$, where $Z_1, \ldots, Z_n$ are the learned high-level causes of $Y$.
2. **Parameterized High-level Models**: The high-level SCM is constrained to a class of linear additive Gaussian noise models $\{\mathcal{H}_\gamma\}_{\gamma \in \Gamma}$ with parameters $\gamma$ to be learned. The exogenous variables $\{W_0, W_1, \ldots, W_n\}$ have a factorized Gaussian distribution $P_W = \prod P_{W_k}$.
3. **Constructive Transformations**: The linear maps $\tau$ and $\omega$ are constructive, meaning each dimension depends only on a designated subset of low-level variables:

$$\tau = (\tau_0, \tau_1, \ldots, \tau_n) \text{ with } \tau_k : x \mapsto \bar{\tau}_k(x_{\pi(k)}) \tag{A.1}$$

$$\omega = (\omega_0, \omega_1, \ldots, \omega_n) \text{ with } \omega_k : i \mapsto \bar{\omega}_k(i_{\pi(k)}) . \tag{A.2}$$

Here, the alignment function $\pi$ partitions the $N$ low-level variables into non-overlapping subsets. Specifically, $\pi(0)$ identifies indices of low-level variables that determine the target $Y$, while $\pi(1), \ldots, \pi(n)$ identify indices for each of the high-level causes. These subsets satisfy $\pi(k) \cap \pi(l) = \emptyset$ for all $k \neq l$, ensuring each low-level variable contributes to at most one high-level variable.

Our non-linear TCR framework may in principle be extended to this multiple cause setting, but it would require the introduction of extra regularization term in the optimized objective in order to constrain the method to separate distinct causes, as elaborated in [10].

## A.3 KL Divergence and Information Geometry

The Kullback-Leibler divergence can be used as a pseudo-metric in the space of probability densities with common support over a given space that we take to be $\mathbb{R}[d]$ for simplicity. By definition, for two densities $p, q$ over the same support

$$D_{\mathrm{KL}}(p\|q) = \int p(x) \log \frac{p(x)}{q(x)} \, \mathrm{d}x .$$

It turns out this can be used to define projections on specific manifolds of probability densities. In particular, one can define the projection of density $p$ on the manifold of Gaussian densities $\mathcal{M}_G$.

$$\mathrm{Proj}_G(p) = \mathcal{N}(\boldsymbol{\mu}(p), \mathbf{Var}(p))$$

where $\boldsymbol{\mu}(p)$ and $\mathbf{Var}(p)$ are the mean and covariance matrix associated to density $p$.

It is easy to check that $\mathrm{Proj}_G(p)$ is the closest point to $p$ on the manifold of Gaussian densities in the sense that

$$D_{\mathrm{KL}}(p, \mathrm{Proj}_G(p)) = \min_{q \in \mathcal{M}_G} D_{\mathrm{KL}}(p, q)$$

This moreover leads to a property analogous to the Pythagorean theorem in Euclidean spaces: for any Gaussian density $q$ and any (possibly non-Gaussian) density $q$ we have the decomposition (see [43] for details)

$$D_{\mathrm{KL}}(p,q) = D_{\mathrm{KL}}(p, \mathrm{Proj}_G(p)) + D_{\mathrm{KL}}(\mathrm{Proj}_G(p), q).$$

While the second term $D_{\mathrm{KL}}(\mathrm{Proj}_G(p), q)$ has a classic analytic expression that is optimized in linear TCR [10], the first term can be written

$$D_{\mathrm{KL}}(p, \mathrm{Proj}_G(p)) = H(\mathrm{Proj}_G(p)) - H(p)$$

where $H(q) = -\int p(x) \log(p(x)) \, \mathrm{d}x$ denotes the entropy associated to density $q$. Using invariance of the KL divergence by parameter transformation, we also have

$$D_{\mathrm{KL}}(p, \mathrm{Proj}_G(p)) = D_{\mathrm{KL}}(p_{\mathrm{norm}}, \mathcal{N}(0, I_d)) = H(\mathcal{N}(0, I_d)) - H(p_{\mathrm{norm}}),$$

where $p_{\mathrm{norm}}$ denotes the density obtain by normalizing the first and second order statistics of $p$ (by removing the mean and dividing by the standard deviation), and $I_d$ denotes the identity matrix for the dimension $d$ of the considered space. This term is called negentropy, as it is up to an additive constant the opposite of the entropy, with the additional specificity that it vanishes when the density is Gaussian (hence with maximum entropy).

The entropy is very challenging to estimate for arbitrary distributions. In contrast, the KL divergence between two (possibly multivariate) Gaussians has an analytic expression.

$$D_{\mathrm{KL}}(\mathcal{N}(\boldsymbol{\mu}_1, \Sigma_1) \| \mathcal{N}(\boldsymbol{\mu}_2, \Sigma_2))$$
$$= \frac{1}{2} \left[ \log(\det(\Sigma_2)/\det(\Sigma_1)) - d + \mathrm{tr}(\Sigma_2^{-1}\Sigma_1) + (\boldsymbol{\mu}_2 - \boldsymbol{\mu}_1)^\top \Sigma_2^{-1}(\boldsymbol{\mu}_2 - \boldsymbol{\mu}_1) \right] \quad \text{(A.3)}$$

Hence, [10] use a Gaussian KL-divergence approximation to optimize linear TCR, defined as

$$D_G(p,q) = D_{\mathrm{KL}}(\mathrm{Proj}_G(p), q). \quad \text{(A.4)}$$

where $q$ is the density of the high-level model, and $p$ is the push-forward density of the low-level model. Because $q$ is assumed Gaussian, this discrepancy has an analytic expression which allows optimizing the loss of Eq. (2.5). From the above development we immediately get that

$$D_G(p,q) \le D_{\mathrm{KL}}(p,q) = D_{\mathrm{KL}}(p, \mathrm{Proj}_G(p)) + D_{\mathrm{KL}}(\mathrm{Proj}_G(p), q). \quad \text{(A.5)}$$

and in particular, the inequality is strict if $p$ is not Gaussian. As a consequence, minimizing $D_G$ does not guarantee a priori that the KL divergence is minimized.

The resulting expression of the consistency loss for linear TCR used by Kekić et al. [10] is thus

$$\mathcal{L}_{\mathrm{cons}}(\tau, \omega, \gamma) = \mathbb{E}_{\boldsymbol{i} \sim P_I} \left[ D_G \left( \widehat{P}_\tau^{(\boldsymbol{i})}(Y, \boldsymbol{Z}) \| P_{\mathcal{H}}^{(\omega(\boldsymbol{i}))}(Y, \boldsymbol{Z}) \right) \right], \quad \text{with } \widehat{P}_\tau^{(\boldsymbol{i})} = \tau_{\#} \left[ P_{\mathcal{L}}^{(\boldsymbol{i})} \right], \quad \text{(A.6)}$$

with the above defined $D_G$.

# B PROOFS OF MAIN TEXT RESULTS

## B.1 UNIQUENESS OF EXACT TRANSFORMATION

**Proposition 4.1** (Uniqueness). *Assume (i) the low-level model is an additive noise SCM of the form*

$$\begin{bmatrix} \boldsymbol{X}_{\pi(1)}, \\ \boldsymbol{X}_{\pi(0)} \end{bmatrix} := \begin{bmatrix} \boldsymbol{f}_1(\boldsymbol{X}_{\pi(1)}) + \boldsymbol{U}_{\pi(1)} + \boldsymbol{i}_{\pi(1)}, \\ \boldsymbol{f}_0(\boldsymbol{X}_{\pi(0)}, \boldsymbol{X}_{\pi(1)}) + \boldsymbol{U}_{\pi(0)} \end{bmatrix}, \quad \boldsymbol{U}_{\pi(1)} \sim P_1 \text{ indep. of } \boldsymbol{U}_{\pi(0)} \sim P_0, \quad \text{(4.2)}$$

*(ii) $P_1$ admits a probability density with non-vanishing Fourier transform, (iii) The high-level model has a non-zero causal effect ($\alpha \ne 0$). Then, if there exists a constructive transformation following Def. 2.3 that is exact for all $\boldsymbol{i}_{\pi(1)} \in \mathbb{R}^{\#\pi(1)}$, it is also unique up to multiplicative and additive constants.*

*Proof.* If there is an exact transformation, marginal probability densities of the target and its corresponding push-forward are matched for all interventions. So the expectations of the target with respect to these densities are matched across interventions, such that

$$\mathbb{E}_{\widehat{P}_\tau^{(i)}}[Y] = \mathbb{E}_{P_{\mathcal{H}}^{(\omega(i))}}[Y],$$

which implies (using linearity of the high-level cause-effect mechanism)

$$\mathbb{E}_{P_{\mathcal{L}}^{(i)}}[\tau_0(\boldsymbol{X}_0)] = \alpha\mathbb{E}_{P(\omega(i))}[Z_1] + \mathbb{E}[W_0]\,.$$

Moreover, the exact transformation assumption also guaranties the match between expectations for the high-level cause, such that the previous equality can be turned into

$$\mathbb{E}_{P_{\mathcal{L}}^{(i)}}[\tau_0(\boldsymbol{X}_0)] = \alpha\mathbb{E}_{P(i)}[\tau_1(\boldsymbol{X}_1)] + \mathbb{E}[W_0]\,.$$

Assuming there exist a solution achieving consistency for all $i$, it must therefore satisfy, up to an additive and multiplicative constant

$$\mathbb{E}_{P(i)}[\tau_1(X_1)] = \mathbb{E}_{P(i)}[\tau_0(X_0)]\,.$$

Consider an alternative solution, it satisfies as well, up to an additive and multiplicative constant

$$\mathbb{E}_{P(i)}[\tilde{\tau}_1(X_1)] = \mathbb{E}_{P(i)}[\tau_0(X_0)]\,.$$

Then for all $\boldsymbol{i}_{\pi(1)} \in \mathbb{R}^{\#\pi(1)}$

$$\alpha\mathbb{E}_{P(i)}[\tilde{\tau}_1(X_1) - \tau_1(X_1)] = 0\,.$$

Assume a non-trivial model, then $\alpha \neq 0$. Using $g = (\mathbf{id} - \boldsymbol{f}_1)^{-1}$, the mapping from exogenous to endogenous variables (restricted to nodes in $\pi(1)$), we can rewrite the condition as

$$\mathbb{E}_{P_1(U-i)}[\tilde{\tau}_1(g(U)) - \tau_1(g(U))] = \int p_1(u-\boldsymbol{i})\delta\tau_1(g(u))\,\mathrm{d}u = p_1 * \delta\tau_1 \circ g(\boldsymbol{i}) = 0\,,$$

where "$*$" denotes the convolution operation. Applying the Fourier transform we get

$$\mathcal{F}\left[p_1 * \delta\tau_1 \circ g\right](\xi) = \mathcal{F}[p_1](\xi) \cdot \mathcal{F}[\delta\tau_1 \circ g](\xi) = 0, \text{ for all } \xi \in \mathbb{R}^{\#\pi(1)}$$

we get that if the Fourier transform of $p_1$ does not vanish, then the Fourier transform of $\delta\tau_1 \circ g$ must vanish everywhere, leading to $\tau_1 \circ g = 0$. Assuming $g$ invertible, we get $\delta\tau_1 = 0$ so the reductions are identical up to an additive and multiplicative constant.

$\square$

## B.2 ANALYTICAL SOLUTION

**Proposition 4.2** (Existence)**.** *Assume a low-level additive SCM following Eq.* (4.2)*. Assume additionally that (i) $U$ is jointly Gaussian, (ii) $\boldsymbol{f}_0(\boldsymbol{X}_{\pi(0)}, \boldsymbol{X}_{\pi(1)}) = \boldsymbol{h}_0(\boldsymbol{X}_{\pi(0)}) + B(\boldsymbol{X}_{\pi(1)} - \boldsymbol{f}_1(\boldsymbol{X}_{\pi(1)}))$, for some functions $\boldsymbol{h}_0, \boldsymbol{f}_1$, some matrix $B \in \mathbb{R}^{(\#\pi(0) \times \#\pi(1))}$, and (iii) $Y = \tau_0(\boldsymbol{X}_{\pi(0)}) = \boldsymbol{a}^\top(\boldsymbol{X}_{\pi(0)} - \boldsymbol{h}_0(\boldsymbol{X}_{\pi(0)}))$ for arbitrary non-vanishing $\boldsymbol{a} \in \mathbb{R}^{\#\pi(0)}$, then the transformation defined by the following maps is exact*

$$\bar{\tau}_1(\boldsymbol{X}_{\pi(1)}) = \boldsymbol{a}^\top B(\boldsymbol{X}_{\pi(1)} - \boldsymbol{f}_1(\boldsymbol{X}_{\pi(1)}))\,, \quad \bar{\omega}_1(\boldsymbol{i}_{\pi(1)}) = \boldsymbol{a}^\top B\boldsymbol{i}_{\pi(1)}\,. \tag{4.3}$$

*Proof.* Because of the choice of $\tau_0$, we have

$$Y = \boldsymbol{a}^\top B\left(\boldsymbol{X}_1 - \boldsymbol{f}_1(\boldsymbol{X}_1)\right) + \boldsymbol{a}^\top\boldsymbol{U}_0 = \boldsymbol{a}^\top B\left(\boldsymbol{U}_1 + \boldsymbol{i}_{\pi(1)}\right) + \boldsymbol{a}^\top\boldsymbol{U}_0\,.$$

Such that for some constant "Cst." independent of $\boldsymbol{i}_{\pi(1)}$

$$\mathbb{E}[Y] = Cst. + \boldsymbol{a}^\top B\boldsymbol{i}_{\pi(1)}\,.$$

Moreover, the high-level model's interventional marginal distribution of $Y$ is given by

$$Y = \alpha Z + W_0 = \tau_1(\boldsymbol{X}_{\pi(1)}^{(0)}) + \alpha\omega_1(\boldsymbol{i}_{\pi(1)}) + W_0 = \alpha\tau_1((\mathbf{id} - \boldsymbol{f}_1)^{-1}(\boldsymbol{U}_1)) + \alpha\omega_1(\boldsymbol{i}_{\pi(1)}) + W_0\,,$$

with $W_0$ an exogenous variable independent of $\tau_1(\boldsymbol{X}_1)$, where we use the map "$(\mathbf{id} - \boldsymbol{f}_1)^{-1}$" from exogenous to endogenous variables of the low-level model restricted to nodes in $\pi(1)$. Note that "$(\mathbf{id} - \boldsymbol{f}_1)$" is invertible by directed acyclic assumption on the graph. If we would allow cycles in the graph, this condition would be obtained by the unique solvability assumption of [42]. By matching

the expectation of $Y$, this implies that up to an arbitrary additive constant $\beta$ (that depends on the choice of $\tau_1$), we have

$$\alpha\omega_1(\boldsymbol{i}_1) = \beta + \boldsymbol{a}^\top B \boldsymbol{i}_{\pi(1)}\,.$$

and we fix $\alpha = 1$ to remove one degree of freedom (because $\tau_1$ and therefore $Z$ can be rescaled accordingly). Now, by cause consistency, we get that the pushforward distribution associated to

$$\tau_1(\boldsymbol{X}^{(\boldsymbol{i})}_{\pi(1)}) = \tau_1((\mathbf{id} - \boldsymbol{f}_1)^{-1}(\boldsymbol{U}_1 + \boldsymbol{i}_{\pi(1)}))\,,$$

matches the shifted high-level distribution of

$$W_1 + \beta + \boldsymbol{a}^\top B \boldsymbol{i}_{\pi(1)}\,.$$

If we reparametrize using $\tau_1(.) = \gamma_1 \circ (\mathbf{id} - \boldsymbol{f}_1)$, we get

$$\gamma_1(\boldsymbol{U}_{\pi(1)} + \boldsymbol{i}_{\pi(1)}) = W_1 + \beta + \boldsymbol{a}^\top B \boldsymbol{i}_{\pi(1)}\,.$$

A Gaussian high-level cause satisfying consistency can thus be achieved by using $\gamma_1 : \boldsymbol{u} \mapsto \beta + \boldsymbol{a}^\top B \boldsymbol{u}$ (because a linear combination of jointly Gaussian random variables is Gaussian, and distribution shifts due to interventions are consistent). Such that we obtain

$$\tau_1 : \boldsymbol{x} \mapsto \beta + \boldsymbol{a}^\top B(\boldsymbol{x}_{\pi(1)} - \boldsymbol{f}_1(\boldsymbol{x}_{\pi(1)}))$$

This is also a sufficient condition to have a valid exact transformation given the Gaussian assumptions.

$\square$

## C  ADDITIONAL METHOD DETAILS

### C.1  DIFFERENTIABLE NORMALITY REGULARIZATION

In Sec. 4.1, we introduced a normality regularization term that encourages the high-level cause distribution to be Gaussian by measuring the Wasserstein distance between the standardized pushforward distribution $\widehat{P}^{(\boldsymbol{i})}_{\tau,\text{std}}(Z)$ and the standard normal distribution $\mathcal{N}(0, 1)$. To implement this regularization in a way that allows end-to-end gradient-based optimization, we need a differentiable approximation of both the empirical cumulative distribution function (CDF) and the distance measure.

Our approach consists of three main steps:

**Standardization.**  Given mini-batches of samples from the pushforward distribution of the high-level cause, we first standardize these samples to have zero mean and unit variance:

$$z_{\text{std}} = \frac{z - \mu_z}{\sigma_z} \tag{C.1}$$

where $\mu_z$ and $\sigma_z$ are the empirical mean and standard deviation of the samples.

**Differentiable CDF Approximation.**  For a differentiable approximation of the empirical CDF, we add smooth step functions of the samples:

$$\hat{F}(x) = \frac{1}{n} \sum_{i=1}^{n} \sigma((x - z_i) \cdot s) \tag{C.2}$$

where $\sigma(t) = \frac{1}{1+e^{-t}}$ is the sigmoid function serving as a smooth version of the step function, $z_i$ are the standardized samples, $n$ is the number of samples, and $s$ is a smoothing factor that controls the sharpness of the transitions in the CDF approximation. Higher values of $s$ create sharper transitions that better approximate the true step function but may lead to sharper gradients.

**Wasserstein Distance Approximation.** We approximate the 1-Wasserstein distance between distributions by computing the $L_1$ distance between their CDFs at a finite set of evaluation points $\{x_1, x_2, \ldots, x_m\}$ uniformly spaced in the interval $[-4, 4]$ (covering most of the probability mass of a standard normal distribution):

$$\hat{W}_1 \approx \frac{1}{m} \sum_{j=1}^{m} |\hat{F}(x_j) - \Phi(x_j)| \tag{C.3}$$

where $\Phi$ is the CDF of the standard normal distribution, computed using the error function: $\Phi(x) = \frac{1}{2}(1 + \mathrm{erf}(x/\sqrt{2}))$.

This approximation gives us a fully differentiable loss function that effectively measures how closely the distribution of our high-level cause follows a Gaussian distribution. In practice, we use the same number of evaluation points as we have samples in our mini-batch, and we set the smoothing factor $s = 10.0$, which provides a good balance between approximation accuracy and gradient stability.

# D ADDITIONAL THEORETICAL RESULTS

As explained in App. A.3, the consistency loss $\mathcal{L}_{cons}$ used in linear TCR [10] is only a KL divergence between the Gaussian approximation of the push-forward distribution of the low-level model and the (Gaussian) high-level model. Therefore, it does not explicitly enforce a perfect match between distributions, which is the theoretical requirement of an exact transformation according to Eq. (2.3).

In principle, however, the setting of Prop. 4.1 does provide additional constraints that theoretically allow identifiability when $\mathcal{L}_{cons}$ vanishes, as shown below.

**Proposition D.1** (Identifiability using $D_G$)**.** *We assume that the prior $P_I$ over interventions has a strictly positive density with respect to the Lebesgue measure. Additionally, we assume the setting of Prop. 4.1: (i) the low-level model is an additive noise SCM of the form*

$$\begin{bmatrix} \boldsymbol{X}_{\pi(1)}, \\ \boldsymbol{X}_{\pi(0)} \end{bmatrix} := \begin{bmatrix} \boldsymbol{f}_1(\boldsymbol{X}_{\pi(1)}) + \boldsymbol{U}_{\pi(1)} + \boldsymbol{i}_{\pi(1)}, \\ \boldsymbol{f}_0(\boldsymbol{X}_{\pi(0)}, \boldsymbol{X}_{\pi(1)}) + \boldsymbol{U}_{\pi(0)} \end{bmatrix}, \quad \boldsymbol{U}_{\pi(1)} \sim P_1 \text{ indep. of } \boldsymbol{U}_{\pi(0)} \sim P_0, \tag{D.1}$$

*(ii) $P_1$ admits a probability density with non-vanishing Fourier transform, (iii) The high-level model has a non-zero causal effect ($\alpha \neq 0$). Then, if there exists a constructive transformation that makes $\mathcal{L}_{cons}$ vanish, it is also unique up to multiplicative and additive constants.*

*Proof.* We notice that the proof of Prop. 4.1 only exploits equality between the expectations of $Z$ or $Y$ obtained under, on the one hand, the high-level model distribution and, on the other hand, the push-forward low-level distribution (for all possible interventions). Since we assume that $\mathcal{L}_{cons}$ vanishes, then the discrepancy $D_G$ must vanish for all $\boldsymbol{i}$[9] and therefore there is a match between the interventional expectations of $\tau_1(\boldsymbol{X})$ for all $\boldsymbol{i}$, and therefore $\tau_1$ is identified following the same argumentation. $\square$

**Potential Limitations to Identifiability based on $\mathcal{L}_{cons}$.** However, the above theoretical identifiability guarantee requires optimizing with respect to an expectation that integrates over all possible values of the intervention vector, which cannot be evaluated in practice, as we necessarily sample from a finite number of interventions. In contrast, linear TCR has theoretical identifiability guarantees under a finite number of interventions [10]. While we could not derive an equivalent result for nonlinear TCR, we conjecture instead a negative result for the loss $\mathcal{L}_{cons}$ of linear TCR that illustrates the practical difficulties of the optimization of $\mathcal{L}_{cons}$.

*Conjecture* D.2. We assume that the prior $P_I$ has a density that vanishes on an open subset of its domain. Additionally, we assume the setting of Prop. 4.1: (i) the low-level model is an additive noise SCM of the form

$$\begin{bmatrix} \boldsymbol{X}_{\pi(1)}, \\ \boldsymbol{X}_{\pi(0)} \end{bmatrix} := \begin{bmatrix} \boldsymbol{f}_1(\boldsymbol{X}_{\pi(1)}) + \boldsymbol{U}_{\pi(1)} + \boldsymbol{i}_{\pi(1)}, \\ \boldsymbol{f}_0(\boldsymbol{X}_{\pi(0)}, \boldsymbol{X}_{\pi(1)}) + \boldsymbol{U}_{\pi(0)} \end{bmatrix}, \quad \boldsymbol{U}_{\pi(1)} \sim P_1 \text{ indep. of } \boldsymbol{U}_{\pi(0)} \sim P_0, \tag{D.2}$$

---

[9]by continuity in $\boldsymbol{i}$ and because its probability density is strictly positive, if the discrepancy was non-zero at any point it would lead to a strictly positive expectation, thus a strictly positive $\mathcal{L}_{cons}$.

(ii) $P_1$ admits a probability density with non-vanishing Fourier transform, (iii) The high-level model has a non-zero causal effect ($\alpha \neq 0$). Then, if there exists a constructive transformation that makes $\mathcal{L}_{\text{cons}}$ vanish, there are infinitely many other constructive transformations that also make $\mathcal{L}_{\text{cons}}$ vanish.

*Justification for the Conjecture.*

Having $\mathcal{L}_{\text{cons}}$ vanish only entails equality between first- and second-order moments of the low-level and high-level push-forward distributions lead to constraints on the convolutions of $\tau_1$ with the exogenous density $p_1$. As exemplified in the proof of Prop. 4.1, using $g = (\mathbf{id} - \boldsymbol{f}_1)^{-1}$ leads to conditions of the form

$$\mathbb{E}_{P(U-\boldsymbol{i})}[\tilde{\tau}_1(g(U)) - \tau_1(g(U))] = \int p_1(u - \boldsymbol{i})\delta\tau_1(g(u))\,\mathrm{d}u = p_1 * \delta\tau_1 \circ g(\boldsymbol{i}) = 0$$

on the support of $P_{\mathbf{I}}$. So we can choose any non-vanishing function $\delta\tau_1$ such that $p_1 * \delta\tau_1 \circ g$ vanishes outside the support of $P_{\mathbf{I}}$ to obtain a different $\tilde{\tau}_1$ satisfying this constraint. This can be done by choosing a function in the Fourier domain of the form $h(\xi) = \mathcal{F}\{p_1\}^{-1}(\xi)\mathcal{F}\{k\}(\xi)$ with $k$ non-zero but vanishing on the support of $P_{\mathbf{I}}$, such that $h$ is $k$ deconvolved by $p_1$. Then we obtain that $p_1 * h = k$, such that $\delta\tau = k \circ g^{-1}$. Given the set of such $k$ is infinite, we conjecture that it is possible to find a class of function satisfying both the constraints on first and second order moments at the same time.

**Practical Issues with the Optimization of $\mathcal{L}_{\text{cons}}$ and Enforcing Non-Gaussianity.** Following the reasoning of the above conjecture, we speculate that optimizing $\mathcal{L}_{\text{cons}}$ with a highly expressive function class may lead to overfitting, in the sense that many highly nonlinear functions can be found that satisfy the first and second order moment constraints of the Gaussian approximation of the KL divergence. Enforcing an exact match between distributions instead of only first and second order moments may help avoid this behavior, *e.g.* by avoiding the nonlinear function to generate multimodal distributions.

While this can be done by adding to $\mathcal{L}_{\text{cons}}$ an addition negentropy term enforcing Gaussianity (see App. A.3), entropy is very challenging to estimate from finite samples and use as a differentiable objective. Instead, we resort to the differentiable measure of non-Gaussianity introduced in Sec. 4.1 based on a Wasserstein distance (see App. C.1) that proved stable in our experiments, and is known to have numerical benefits over the KL divergence.

Experiments using this measure as a regularizer are shown in App. F.1 and are compatible with our theoretical insights, in the sense that it improve performance when the sampling of the intervention does not cover a broad enough range of the exogenous distribution.

# E EXPERIMENTAL DETAILS

**Data.** For each experiment we sample $N_{\text{int}}$ distinct interventions and for each distinct intervention, we generate $N_{\text{ep}}$ RL episodes that apply that intervention. For the synthetic experiments, we equivalently sample $N_{\text{ep}}$ SCM samples that apply the same intervention. That is, we keep the intervention fixed but re-sample the exogenous noise. For all datasets, we use $80\%$ for training and $10\%$ for validation and test, respectively.

## E.1 SYNTHETIC EXPERIMENTS

**Nonlinear Causal Model for Synthetic Data.** For our synthetic experiments, we implement a class of nonlinear causal models that follow the structure described in Prop. 4.2. That is, we sample SCMs with the structure given in Eq. (4.2), where, $|\pi(0)| = 2$ and $|\pi(1)| = 9$, creating an 11-dimensional system (9 low-level cause variables plus 2 target variables). The vector $\boldsymbol{a} \in \mathbb{R}^2$ and the matrix $B \in \mathbb{R}^{1 \times 9}$ have entries sampled from a standard normal distribution, providing the linear mapping from causes to effect and from the low-level variables in $\pi(0)$ to the target, respectively, as required by Prop. 4.2. The exogenous noise variables $\boldsymbol{U}_{\pi(1)}$ and $\boldsymbol{U}_{\pi(0)}$ are sampled from a normal distribution with zero mean and standard deviation 1.0. The nonlinear functions $\boldsymbol{f}_1$ and $\boldsymbol{h}_0$ consist of component-wise nonlinear mechanisms represented by randomly initialized two-layer neural networks with 10 hidden units and `tanh` activation functions.

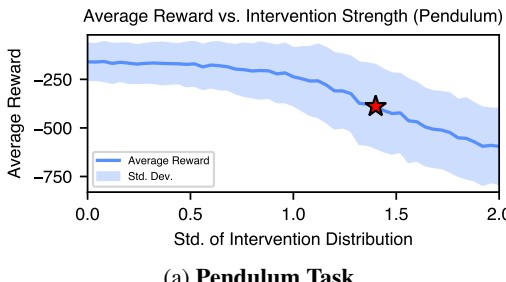 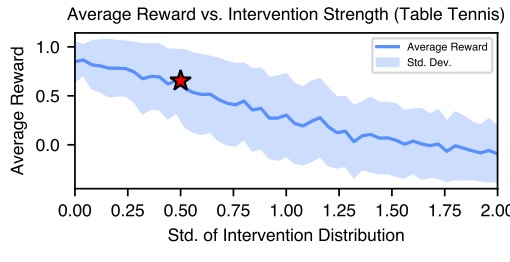

(a) **Pendulum Task**           (b) **Table Tennis Task**

Figure A: **Relationship Between Intervention Strength and Task Performance.** These plots illustrate how the average episode reward varies as a function of the intervention strength $\sigma$ (standard deviation of the Gaussian perturbations $\delta A_t \sim \mathcal{N}(0, \sigma^2)$) for both RL tasks. The shaded regions represent $\pm 1$ standard deviation of the reward distribution across episodes. The star markers indicate our selected intervention strengths, which were chosen at the threshold where increased perturbation begins to significantly degrade policy performance. This selection balances between introducing sufficient causal signal for the reduction algorithm and preserving the policy's core behavior patterns.

**Intervention Generation.** For each experiment, we generate interventions by sampling from a standard normal distribution, creating shift interventions $i_{\pi(1)} \sim \mathcal{N}(0, \mathbb{1})$. We generate $N_{\text{int}} = 10^7$ distinct interventions, and for each intervention, we sample $N_{\text{ep}} = 1024$ data points from the causal model.

**Neural Network Architecture for nTCR.** For these synthetic data experiments, the reduction maps $\tau_1$ and $\omega_1$ are parameterized using neural networks with residual connections. Each network consists of a first linear layer that maps the input dimension to a hidden dimension of 256, followed by 8 hidden layers with Softplus activation functions and residual connections. The final layer maps the hidden representation to the high-level cause/intervention. This architecture with residual connections helps in learning complex functions while maintaining gradient flow during training. This amounts to approximately $1.2 \cdot 10^7$ parameters for $\tau_1, \omega_1$ and the high-level causal model combined. Unlike our RL experiments, we do not use the Gaussian kernel representation described in Sec. 4.3 since we are only interested in verifying the identification properties of our approach, not interpretability.

**nTCR Training.** We train our nTCR model using the Adam optimizer with an initial learning rate of $5 \cdot 10^{-4}$ and a cosine annealing learning rate scheduler that gradually reduces the learning rate to zero over the course of training. For the normality regularization, we set $\eta_{\text{norm}} = 1.0$ to encourage the learned high-level cause to follow a Gaussian distribution. We use a batch size of 512, meaning that each training step processes 512 distinct interventions sampled from our prior distribution $P(i)$.

**Identification Metric.** To quantify how well our learned reduction maps align with the ground truth solutions from Prop. 4.2, we employ a normalized, debiased L2 loss. Let $z_i^{\text{learned}} = \tau_1^{\text{learned}}(X_i)$ be the high-level cause computed using our learned reduction, and $z_i^{\text{gt}} = \tau_1^{\text{gt}}(X_i)$ be the corresponding ground truth value for sample $i$. We first standardize both sets of samples:

$$\tilde{z}_i^{\text{learned}} = \frac{z_i^{\text{learned}} - \overline{z^{\text{learned}}}}{\sqrt{\sum_j (z_j^{\text{learned}} - \overline{z^{\text{learned}}})^2}} \tag{E.1}$$

$$\tilde{z}_i^{\text{gt}} = \frac{z_i^{\text{gt}} - \overline{z^{\text{gt}}}}{\sqrt{\sum_j (z_j^{\text{gt}} - \overline{z^{\text{gt}}})^2}} \tag{E.2}$$

where $\overline{z^{\text{learned}}}$ and $\overline{z^{\text{gt}}}$ are the respective sample means. We then find the optimal rescaling coefficient $c$ that minimizes the L2 distance:

$$c = \arg\min_c \sum_i (c \cdot \tilde{z}_i^{\text{learned}} - \tilde{z}_i^{\text{gt}})^2 = \frac{\sum_i \tilde{z}_i^{\text{learned}} \cdot \tilde{z}_i^{\text{gt}}}{\sum_i (\tilde{z}_i^{\text{learned}})^2} \tag{E.3}$$

The final identification loss is the normalized minimum L2 distance:

$$\mathcal{L}_{\text{id}}(\tau) = \frac{1}{\sqrt{n}} \sqrt{\sum_i (c \cdot \tilde{z}_i^{\text{learned}} - \tilde{z}_i^{\text{gt}})^2} \tag{E.4}$$

where $n$ is the number of samples. The same procedure is applied to evaluate the $\omega_1$ map. This metric accounts for the fact that the theoretical solutions are only identified up to multiplicative and additive constants, while measuring how well the learned reductions capture the structural properties of the ground truth solutions.

**Compute Resources.** All synthetic experiments were conducted on NVIDIA Quadro RTX 6000 GPUs with 4 CPU cores allocated per run. Each complete training run required approximately 48 hours of computation time. We performed 10 different runs, each with a different randomly generated causal model.

## E.2 PENDULUM

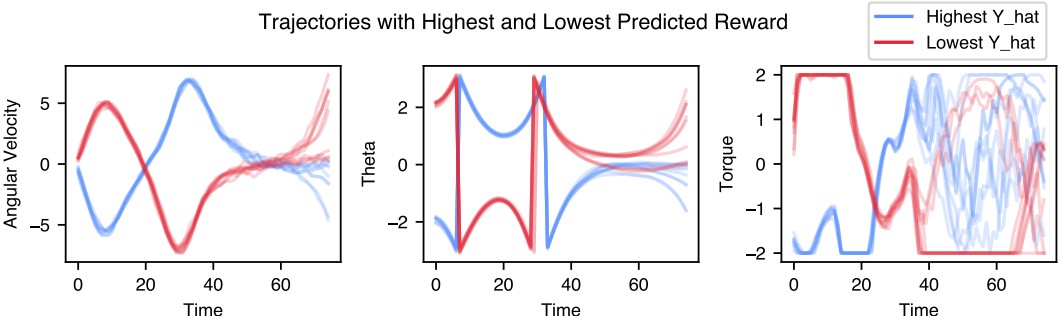

Figure B: **Episodes with Highest and Lowest Predicted Reward.** The plot shows episodes from the test dataset with the highest and lowest cumulative reward predicted by nTCR for Policy A. The red lines display the 10 episodes with highest predicted rewards, while the blue lines show the 10 episodes with lowest predicted rewards.

Each episode in the Pendulum task [21] has a length of $T = 200$ steps. We apply interventions to the actions for the first 75 steps, and define the target variable as the cumulative reward for the remainder of the episode $Y = \sum_{t=76}^{200} R_t$. The reward of the Pendulum task is defined as $R_t = -\theta_{t-1}^2 - 0.1\,\dot{\theta}_{t-1}^2 - 0.001\,A_t^2$, which penalizes deviations from the upright position, large angular velocities, and large torque applied by the policy.

**RL Agent Training.** We train the policies for Sec. 5.2 with the Stable-Baselines 3 [44] implementation of Proximal Policy Optimization (PPO) [45]. For the Pendulum experiments, we use the default hyperparameters of Stable-Baselines 3. The RL agent collects 1 million environment transitions, and the total wall-clock training time is 50 minutes on an Intel Xeon W-2245 CPU. The training requires less than 1 GB of memory.

**Policy Data.** For the analysis of the Pendulum policy, we collect a dataset of $N_{\text{int}} = 100{,}000$ interventions with $N_{\text{ep}} = 100$ episodes per intervention. Collecting the datasets of Policy A and B in Sec. 5.2 takes about 150 CPU core-hours each on a compute cluster, with 4GB of memory per core.

**nTCR.** We train our nTCR model using the Adam optimizer with an initial learning rate of $0.01$ and a cosine annealing learning rate scheduler that reduces the learning rate to zero over the course of training. We use a batch size of $64$ and train for $90$ epochs with weight decay of $0.01$. For the normality regularization, we set $\eta_{\text{norm}} = 10$.

For our interpretable nonlinear function class (Sec. 4.3), we use $128$ Gaussian kernels for each variable at each time step. Before training, we determine the minimum and maximum values $x_{\min}^j$

and $x^j_{\max}$ for each variable $j$ across all episodes. The kernel centers $\{\mu_{j,t,k}\}^{128}_{k=1}$ are equally spaced over the range $[x^j_{\min}, x^j_{\max}]$, and the kernel widths are set as:

$$\sigma_{j,t} = c \cdot \frac{x^j_{\max} - x^j_{\min}}{128} \tag{E.5}$$

where the constant $c = 8$ acts as a smoothing parameter—higher values lead to smoother learned reduction functions.

For computational convenience, we pre-sample all episodes under interventions and store the data rather than simulating episodes during training, though the latter approach would also be feasible in principle.

**Episodes with High and Low Predicted Reward.** Fig. B displays episodes from the test set with the highest and lowest predicted cumulative rewards according to the learned nTCR model for Policy A. These episodes correspond to the characteristic trajectory patterns identified in Fig. 3(left): clockwise swinging motions starting from the right quadrant tend to have higher rewards than episodes with counterclockwise motion originating from the left quadrant.

**Compute Resources for nTCR.** All nTCR training experiments for the Pendulum task were conducted on NVIDIA Quadro RTX 6000 GPUs with 8 CPU cores allocated per run. Each complete training run required approximately 2 hours of computation time and used roughly 4 GB of GPU memory.

### E.3 TABLE TENNIS

In the simulated table tennis task [22, 23], episodes terminate when the racket hits the ball, which typically happens between steps 90 and 100. The environment then continues simulating the ball to determine its landing position. If the agent does not hit the ball, the episode terminates as soon as the ball falls below the height of the table, which typically happens after 125 to 135 steps. Since our method requires episodes of fixed length, we pad the data to $T = 150$ steps by appending zeros after episode termination.

The agent receives a non-zero reward only at the end of the episode, which depends on whether the agent managed to hit the ball. This reward is given by

$$R_t = \begin{cases} R^{\text{tt}} & \text{racket touches the ball} \\ R^{\text{hit}} & \text{ball below the table} \\ 0 & \text{otherwise}, \end{cases} \tag{E.6}$$

where $R^{\text{tt}}$ rewards the agent for hitting the ball close to the desired position on the table and $R^{\text{hit}}$ penalizes missing the ball depending on the minimum distance between the ball and the racket. These reward terms are defined as

$$R^{\text{tt}} = \max\left(1 - \left(\frac{\|\boldsymbol{p}^{\text{land}} - \boldsymbol{p}^{\text{des}}\|}{3}\right)^{\frac{3}{4}}, -0.2\right) \tag{E.7}$$

$$R^{\text{hit}} = -\min_t \|\boldsymbol{p}^{\text{ball}}_t - \boldsymbol{p}^{\text{racket}}_t\|_2, \tag{E.8}$$

where $\boldsymbol{p}^{\text{land}}$ is the landing point of the ball on the table, $\boldsymbol{p}^{\text{des}}$ is the desired landing point at the center of the opponent's side, and $\boldsymbol{p}^{\text{ball}}_t$ and $\boldsymbol{p}^{\text{racket}}_t$ are the 3D positions of ball and racket at timestep $t$, respectively.

**RL Agent Training.** Similar to the Pendulum task, we train the table tennis policies in Sec. 5.3 with the Stable-Baselines 3 [44] implementation of PPO [45]. We use the algorithm hyperparameters tuned by Guist et al. [23], which are displayed in Tab. A. The RL agent collects 3 million environment transitions, and the total wall-clock training time is roughly 10 hours on an Intel Xeon W-2245 CPU. Running the simulation and the RL training requires 8 GB of memory.

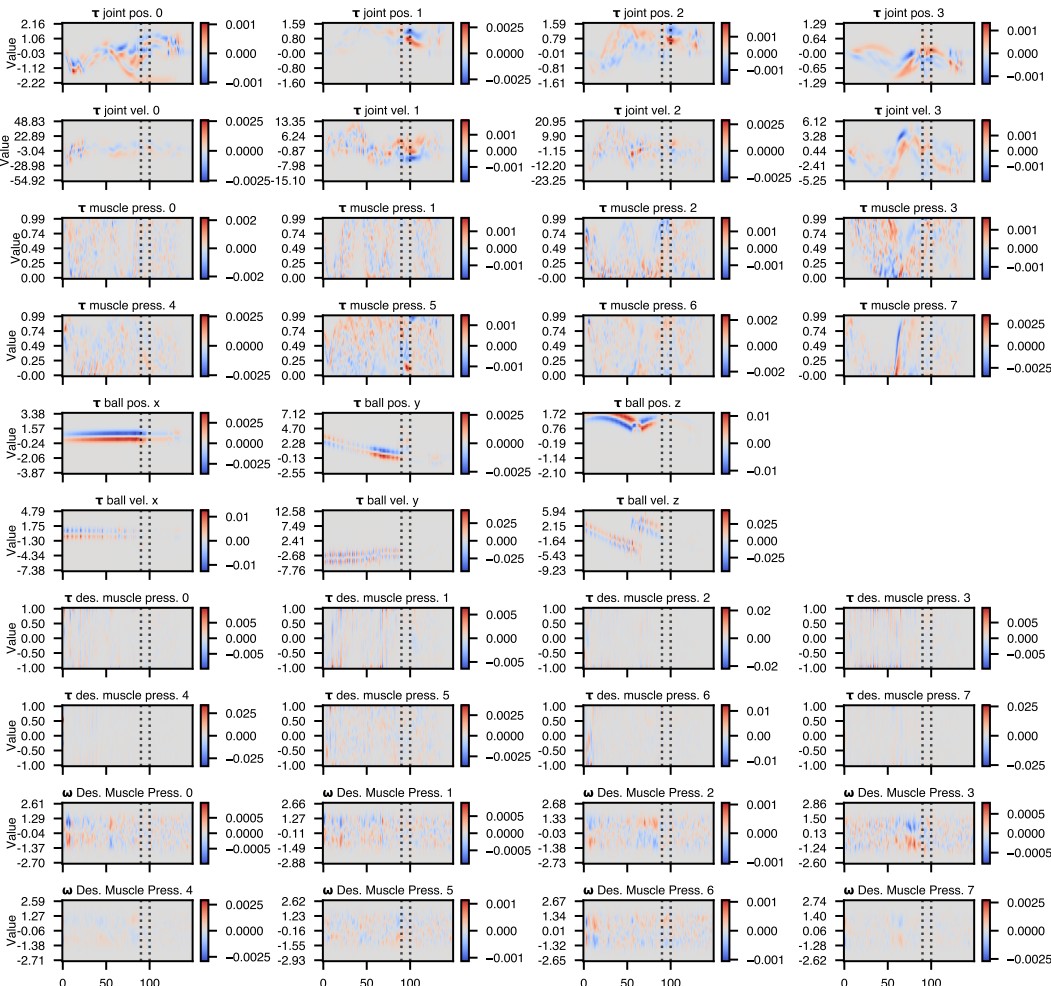

Figure C: **nTCR Reduction Maps for Table Tennis Task.** Linear parameter in high-level model 0.0013 (bias 0.5964).

**Policy Data.** For the table tennis analysis, collecting the dataset of $N_{\text{int}} = 100,000$ interventions with $N_{\text{ep}} = 100$ episodes per intervention takes about 3400 CPU core-hours, with 8GB of memory per core.

**nTCR.** We use similar nTCR training settings as for the Pendulum task, with the following modifications: learning rate of 0.0001 and weight decay of 0.1 (manually tuned for optimization stability), 32 Gaussian kernels per variable-time pair (chosen to fit GPU memory constraints), and smoothing parameter $c = 1$ (smaller than Pendulum to provide necessary resolution and avoid oversmoothing with fewer kernels). Fig. C shows the full set of nTCR maps for the table tennis task.

**Compute Resources for nTCR.** All nTCR training experiments for the table tennis task were conducted on NVIDIA A100-SXM4-40GB GPUs with 8 CPU cores allocated per run. Each complete training run required approximately 5 hours of computation time and used around 20 GB of GPU memory.

Table A: **Stable-Baselines 3 PPO Hyperparameters used for Training the Table Tennis Policy.**

| Parameter name | Value |
| --- | --- |
| batch_size | 310 |
| clip_range | 0.4 |
| clip_range_vf | 0.3 |
| ent_coef | $7 \times 10^{-6}$ |
| gae_lambda | 1 |
| gamma | 0.9999 |
| learning_rate | 0.00015 |
| max_grad_norm | 0.1 |
| n_epochs | 34 |
| n_steps | 5000 |
| num_hidden | 380 |
| num_layers | 1 |
| vf_coef | 0.56 |

# F ADDITIONAL EXPERIMENTAL RESULTS

## F.1 EFFECT OF NORMALITY REGULARIZATION

To better understand the role of our normality regularization introduced in Sec. 4.1, we conduct an ablation study on synthetic data across different noise regimes. We vary the variance of the intervention distribution while keeping the exogenous noise levels in the low-level SCM constant, effectively creating different signal-to-noise ratios.

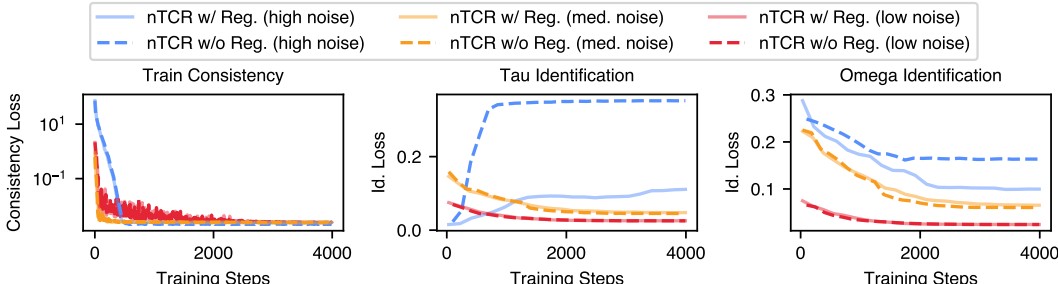

Figure D: **Effect of Normality Regularization on Achieved Consistency and Identification of Ground-Truth Solution.** The figure shows the consistency loss (left) and the identification losses measuring agreement with the ground-truth solutions (definition in App. E.1) for the $\tau$- and $\omega$-functions (middle and right) over the reduction training run for synthetic low-level SCMs with 3 low-level variables. The interventions are sampled as $\boldsymbol{i}_{\pi(1)} \sim \mathcal{N}(0, h \cdot \mathbb{1})$, with $h \in \{0.1, 1.0, 10.0\}$ for the high-, medium- and low-exogenous-noise settings, respectively. Note that, since we keep the exogenous noise levels in the low-level SCM constant, a low variance of the intervention distribution corresponds to high exogenous noise relative to the interventions. The lines for normality regularization have $\eta_{\text{norm}} = 1$; to turn off regularization we set $\eta_{\text{norm}} = 0$. We show average values over 10 sampled SCMs.

Fig. D shows several important insights about the effect of normality regularization across different noise regimes. First, the identifiability metrics demonstrate that learning ground truth reduction maps becomes progressively more challenging as we move from the low-noise setting (high intervention variance) to the high-noise setting (low intervention variance). This reflects the fundamental signal-to-noise ratio: when interventions have small variance relative to the exogenous noise, the signal about the most influential factors for the target becomes harder to detect and extract.

Most significantly, we observe that normality regularization has the strongest beneficial effect in the high-noise regime, where it substantially improves the identification of ground truth solutions for

both $\tau$ and $\omega$ maps. In contrast, for the low- and medium-noise settings, the regularization provides minimal additional benefit, suggesting that when the signal-to-noise ratio is favorable, the consistency loss alone is sufficient to guide the learning toward the correct solution. To understand why normality

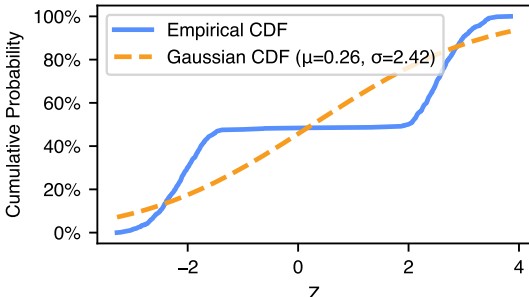

Figure E: **Non-Gaussianity of the High-level Cause.** The figure shows the empirical CDF (blue) for the high-level cause for an SCM sampled in Fig. E (high noise setting) for an nTCR with $\eta_{\mathrm{norm}} = 0$. The yellow dashed line shows the CDF for a Gaussian distribution with the same mean and variance.

regularization is particularly crucial in the high-noise regime, Fig. E shows the empirical CDF of the learned high-level cause distribution in the high-noise setting without normality regularization. The distribution shows highly non-Gaussian behavior with multiple peaks, indicating that the $\tau$ reduction is exploiting the flexibility of the nonlinear function class to minimize the consistency loss by artificially shaping the high-level pushforward distribution. Such multi-modal distributions are undesirable for interpretability, as they introduce additional complexity that obscures the underlying causal structure (see also our discussion of Gaussianity in App. D).

These results demonstrate that normality regularization serves as an important inductive bias that prevents overfitting to the training data while maintaining the interpretability of the learned high-level causes. The regularization ensures that interventions must be sufficiently strong relative to the exogenous noise to be effectively learned - a principle that aligns with the fundamental requirement that causal interventions should produce detectable changes in the target phenomenon (see also Fig. A).

## F.2 LINEAR TCR FOR PENDULUM POLICIES

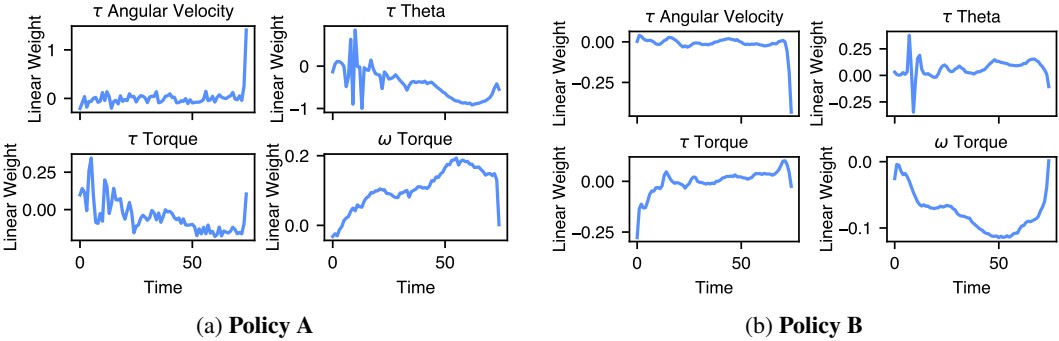

(a) **Policy A**  (b) **Policy B**

Figure F: **Linear TCR [10] for Pendulum Task.** Linear parameters in high-level model: $5.6821$ (bias $-28.5229$) for Policy A and $94.0380$ (bias $-115.0152$) for Policy B.

The reduction maps for linear TCR [10] for policies A and B are shown in Figs. Fa and Fb, respectively. These provide a baseline for comparison with our nonlinear approach (nTCR).

**Policy A: Limitations of Linear TCR.** For Policy A, linear TCR identifies predominantly negative weights in the second half of the episode for the $\tau$-maps for Theta and Torque. While this correctly captures that deviations from the upright position (nonzero Theta) and large corrective actions (high Torque) late in episodes correlate with lower rewards, the linear approach fails to capture the

more nuanced pattern revealed by nTCR. For the first half of the episode, the linear weights show inconsistent, unstable patterns without a clear interpretable structure. The $\tau$-map for Angular Velocity appears largely neutral except for an unexplained positive peak at the final time step.

In contrast, nTCR (as shown in Fig. 3, left) clearly distinguishes between two distinct trajectory classes: clockwise swinging (from the right quadrant) versus counterclockwise swinging (from the left quadrant). This critical asymmetry in policy performance is entirely missed by the linear model, which can only capture monotonic relationships between state variables and expected reward.

**Policy B: Observations Consistent with nTCR.**   For Policy B, linear TCR shows a strongly negative $\tau$-map for Torque at the start of episodes and a relatively neutral pattern in the second half. This suggests episodes starting with positive torque tend to have worse outcomes. The $\omega$-map indicates shifting torque toward more negative values would be beneficial, which aligns with nTCR's findings. These two observations are consistent with Policy B's failure mode—allowing the pendulum to tip over in the positive $\theta$ direction before applying corrective torque—can be captured by linear relationships. In this case, both linear TCR and nTCR correctly identify the key intervention (applying more negative torque) that would improve policy performance.

**Comparative Advantages of nTCR.**   Overall, we observe that linear TCR has fundamental limitations in capturing complex behavioral patterns in RL policies:

- **Non-monotonic relationships:** Linear TCR cannot capture U-shaped or other non-monotonic relationships between states and expected rewards.
- **Trajectory classes:** Linear TCR fails to distinguish between qualitatively different trajectory classes (like clockwise vs. counterclockwise motion) that are not linearly separable.

These results empirically validate the need for our nonlinear extension to TCR, demonstrating that nTCR can uncover important qualitative patterns in policy behavior that remain hidden to linear approaches.

### F.3   LINEAR TCR FOR TABLE TENNIS POLICIES

The reduction maps for linear TCR [10] for the table tennis task are shown in Fig. G. While linear TCR provides a computationally simpler baseline, it reveals fundamental limitations when applied to complex robotic control tasks.

**Temporal Importance Identification.**   Linear TCR successfully identifies that the most crucial periods for policy success or failure occur just before and during ball contact. This can be observed from the largest magnitude contributions in the linear reduction maps, which appear predominantly around the time window when the robot typically hits the ball (indicated by the dotted vertical lines). This temporal insight aligns with the physical intuition that precise timing and positioning during ball contact are critical for successful returns.

**Limited State Representation.**   However, the resolution of linear TCR is fundamentally constrained: it can only indicate whether a particular variable at a specific time contributes positively or negatively to the expected reward *on average* over the entire state and action distribution observed in the dataset. Linear reductions cannot distinguish between different values that a variable might take at any given time step. For instance, while nTCR can identify that balls bouncing closer to the net present greater challenges for the robot (Fig. 4(h,i)), it would be hard to deduce such value-dependent relationships from linear TCR. The linear approach averages over all ball positions at each time step, obscuring the specific spatial patterns that influence task difficulty.

**Expressivity vs. Simplicity Trade-off.**   While linear TCR offers benefits in terms of training simplicity and straightforward interpretation of the learned reductions, its limited capacity to represent state-dependent relationships severely constrains its applicability to complex control scenarios. The table tennis task exemplifies this limitation: successful policy explanation requires understanding not just *when* certain variables matter, but also *which specific values* of those variables lead to success or failure. This distinction between temporal importance and state-value dependencies shows the necessity of our nonlinear extension for capturing meaningful causal patterns in more complex tasks.

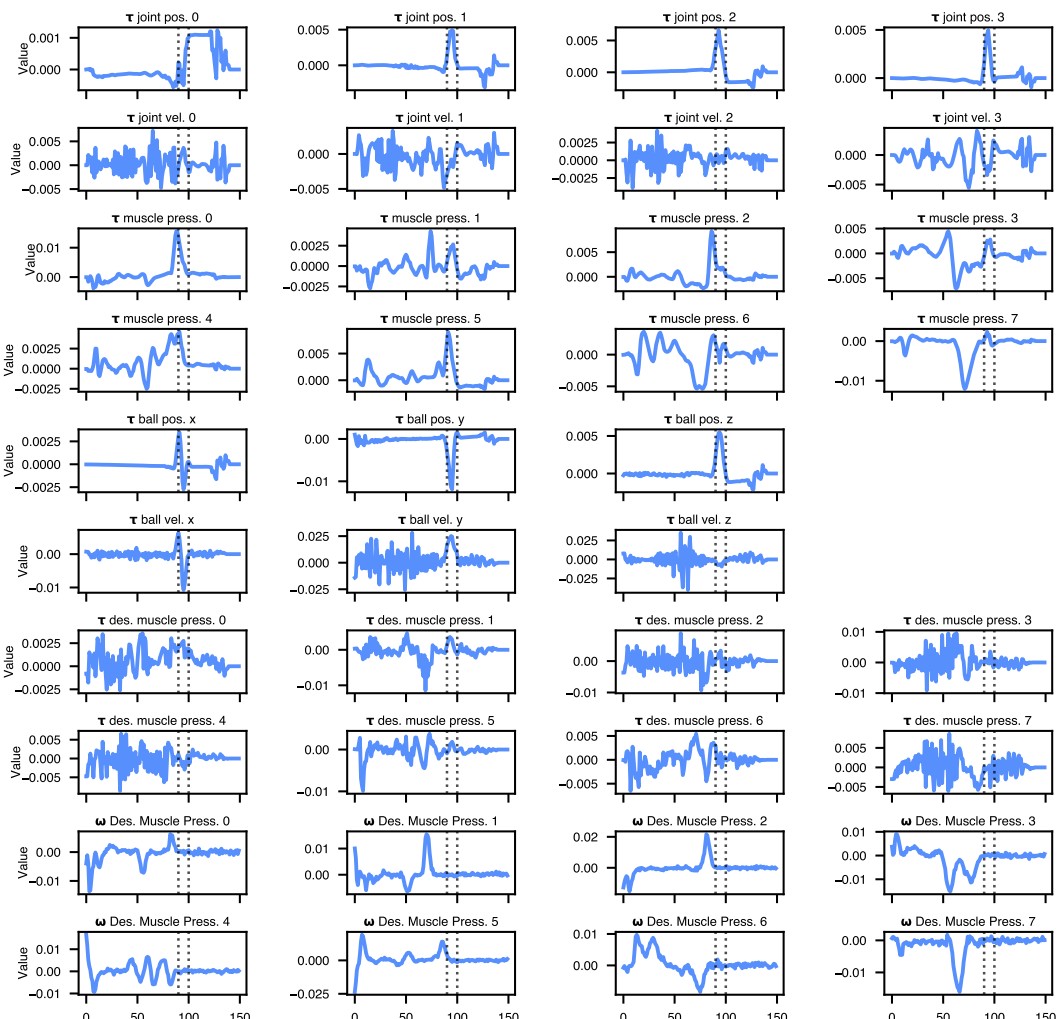

Figure G: **Linear Reduction Maps for Table Tennis Task.** Linear parameter in high-level model 0.6933 (bias 0.6780).

## F.4    COMPARISON OF CONSISTENCY LOSS BETWEEN LINEAR AND NONLINEAR TCR

## F.5    PENDULUM POLICY REDUCTIONS UNDER DIFFERENT INTERVENTION STRENGTHS

Fig. I shown our nTCR approach across different intervention strengths for Policy A of the Pendulum task presented in Sec. 5.2. We vary the standard deviation $\sigma$ of the Gaussian perturbations applied to the policy's actions from very weak ($\sigma = 0.1$) to very strong ($\sigma = 3.0$). The intervention strength in the main experiments in Fig. 3 were chosen heuristically: the interventions were chosen to be strong enough to induce a slight but measurable deterioration in the average reward.

For weak interventions ($\sigma = 0.1$), the reduction maps are hard to learn, as the perturbations are insufficient to generate measurable variations in cumulative rewards. While there is a resemblance to the patterns in Fig. 3, the low signal-to-noise ratio prevents reliable identification of the behavioral structure. As we increase the intervention strength to moderate levels ($\sigma = 1.2$ to $1.6$), clear and consistent patterns emerge. The distinctive clockwise versus counterclockwise trajectory separation that characterizes Policy A's bias becomes clearly visible across this range. The $\omega$ maps also remain qualitatively stable in this regime, with the magnitude of the map increasing with increasing range of observed interventions.

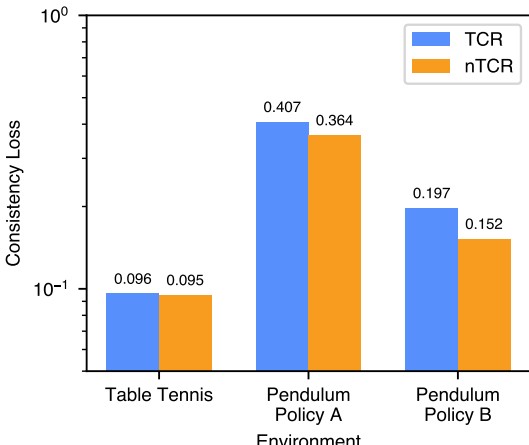

Figure H: **Consistency Loss Comparison.** The figure shows the interventional consistency loss (2.5) achieved by linear TCR (blue) and nonlinear TCR (orange) on validation data for the three policies shown in Sec. 5.

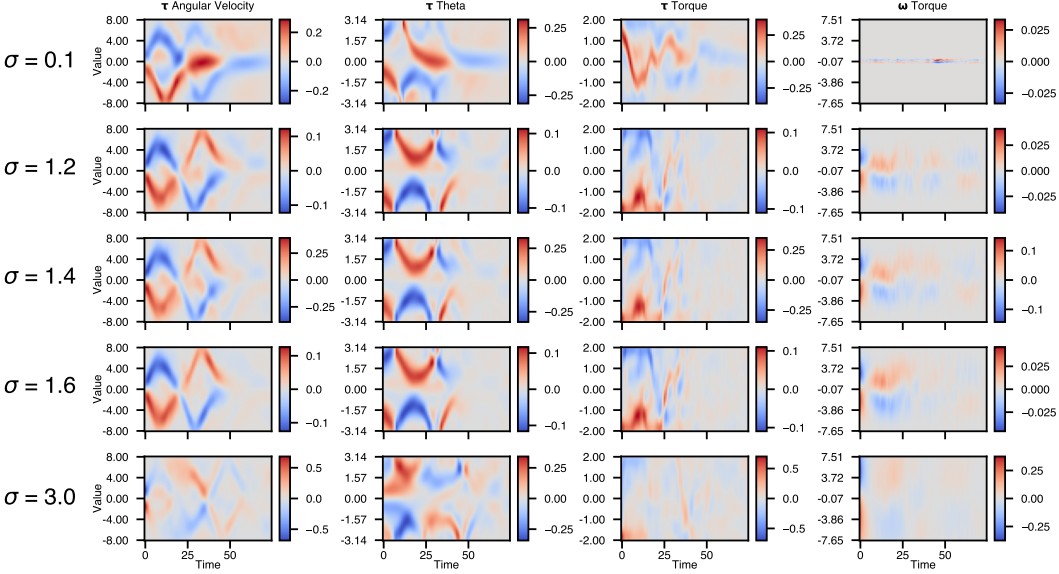

Figure I: **Pendulum nTCR under increasing Intervention Strength.** The figure shows nTCR maps for Policy A shown in Sec. 5.2 for different intervention strengths $\sigma$. $\sigma = 1.4$ is also shown in Fig. 3.

When interventions become too strong ($\sigma = 3.0$), the learned patterns begin to diffuse and lose sharpness. This occurs because large perturbations push the policy far outside its normal regime, generating trajectories that no longer reflect the policy's intended behavior. Despite this degradation at extreme values, our key findings about Policy A's rotational bias remain stable across a practical range of intervention strengths ($\sigma \in [1.2, 1.6]$), demonstrating the robustness of our approach to this hyperparameter.

## G ADDITIONAL RELATED WORK

While extensive theoretical work has established the foundations of causal abstractions [13–16, 19], practical applications of these frameworks have only recently begun to emerge. Notable applications include understanding the behavior of language models [17, 35] and neural networks more generally [46], and developing causally-informed approaches to multi-armed bandits [47]. These emerging

applications demonstrate the growing practical relevance of causal abstraction theory across many domains.

Notions of abstraction, with the idea of reducing the dimensionality of a system, have also been explored in RL. Temporal abstraction [48] reduces the time dimension, while state abstraction [49] reduces the set of states of a Markov Decision Process (MDP). Both types of RL abstraction share the characteristic of preserving an MDP-like structure by only exchanging fine-grained states or actions for coarse grained-ones. Both approaches have been primarily used to learn more efficient and robust RL policies by reducing the dimensionality of the problem, but have also been explored to provide explanations. A notable examples is [31] which learns a graph between abstract states to explain how a policy works.

In contrast, TCR eliminates the MDP structure to focus on explaining a single target variable (a final/cumulative reward in RL) and a few direct causes and interventions on them. The explanation provided is then of a different nature, focusing on "why" a policy succeeds or fails (explaining outcomes) in relation to a high-level description of the relevant aspects ("what") of the system it is acting on, instead of explaining its decisions and therefore "how" it operates. Moreover, TCR is applicable to any learned policies (it does not put constrained on the training) and allows for comparison between policies based on fundamentally different approaches.

Another type of explanation in RL are counterfactuals or "what-if" scenarios. For instance, [29] builds interpretable parametrizations of sequential decision-making by explaining a policy in terms of preferences over "what-if" outcomes. In [50] a search algorithm for optimal counterfactual explanations is presented. The counterfactual explanation specifies an alternative sequence of actions differing in at most k actions from the observed sequence that could have led the observed process realization to a better outcome. In contrast to counterfactual approaches, and more generally to explanation of a single trajectory/data-point, TCR provides a coarse-grained overview of the key factors that alter the outcomes, on average across all scenarios.

## H    USE OF LARGE LANGUAGE MODELS (LLMs)

LLMs were used to polish the writing of the paper.

