# OpenReview forum: "Learning Nonlinear Causal Reductions to Explain Reinforcement Learning Policies"
_ICLR.cc/2026/Conference — ICLR 2026 Poster_

### Official Review · Reviewer_heb9 · 2025-10-31

**Soundness:** 3
**Presentation:** 3
**Contribution:** 2
**Rating:** 4
**Confidence:** 4

**Summary:**

The paper presents an extension of the causal model reduction framework to better address the problem of explaining RL policies. Previous work in this area only used linear models. The authors extended this by introducing a class of non-linear reduction functions that remain interpretable. This can better capture behaviors in the learned reductions and help explain policies. The paper provides theoretical guarantees of the correctness of their approach, along with an experimental evaluation that demonstrates that the approach uncovers patterns leading to target behaviors in two RL tasks.

**Strengths:**

- The problem is well motivated and relevant.
- The theoretical contribution seems rigorous.

**Weaknesses:**

- The writing in the paper is quite dense in several areas, specifically in sections 4 and 5.
- Lack of comparison with similar approaches for causal reduction.
- Evaluation is limited to specific use cases/configurations.

**Questions:**

At this stage, my initial recommendation is to reject the paper, although this is not a strong or definitive rejection. My main concern with the submission is the evaluation, which I find insufficient to convincingly support the paper's claims. There is no sufficient evaluation comparing previous approaches and the suggested one. The only comparison available is on synthetic data with specific parameters, which provides limited evidence for the paper's contribution. Also, the writing is somewhat dense in several sections of the paper, particularly in sections 4 and 5, but this part can be improved with a small revision.

Detailed questions and comments to the authors:

- Is the use of linear targeted causal reduction for explaining RL agents discussed in previous work? This part of the paper should be clearer.

- In line 261, what is $\eta_{norm}$? I could not find an explanation in the text.

- Could you provide some intuition about why the assumptions made in Propositions 4.1 and 4.2 are reasonable within the task you aim to address? I did not find a discussion about this in the paper.

- It would be great if the authors could provide more details about the choice of function described in Section 4.3. Specifically, are these functions used in previous work for similar cases? Can they capture linear behavior patterns? How are the fixed parameters set?

- Section 5.1: Could you comment on the choice of configuration you made for this validation? Most importantly, setting the h function to zero. I like the idea of using synthetic data for validation, but the evaluation seems limited to very specific parameters, which narrows the authors' claim about validation. This is especially unusual, given that it is synthetic data, which can be relatively easy to generate and experiment with.

- sections 5.2-5.3: The purpose of this evaluation is somewhat unclear. I was expecting a comparison between TCR and nTCR, focusing on accuracy (I guess sampled from the environment since we lack ground truth) and the computational overhead each one requires. It would help if the authors could clarify what they aimed to evaluate.

- Although the pendulum and table tennis are interesting and well known use cases, they contain a low number of variables (especially the pendulum). This makes me think, can you comment on the limitations of the current reduction approaches? Is a reduction necessary in cases like explaining a pendulum agent?

Additional feedback:

I found sections 4 and 5 to be a bit dense. It involves a lot of technical details without enough explanation or context. I recommend adding explanations or interpretations of the main points for the terms and choices discussed in Section 4, as well as the details of the experiments in Section 5.You can allocate space for this by shortening the beginning of Section 2 or using the extra page provided if the paper is accepted.

---

> ### Author Response · Authors · 2025-11-22
> **Response to Reviewer heb9 (1/3)**
>
> We thank the reviewer for their detailed review and helpful suggestions for improving our work. We appreciate the time and effort invested in reviewing our work, and we're pleased that you found our problem well-motivated and our theoretical contribution rigorous. Below, we address your main concerns and questions.
>
> ## Synthetic Experiments
>
> The function $h_0$ represents the causal relationships among the low-level variables in $\pi(0)$ before they are mapped to the target $Y$. Since we set $|\pi(0)| = 1$, there are no inter-variable relationships to model, making $h_0$ trivially zero.
> You raise a good point about testing the more general case. We have updated the experiments in Section 5.1 to use $|\pi(0)| = 2$ with nontrivial $h_0$ functions sampled from the same neural network class used for the other causal mechanisms.
> The results shown in Figure 2 demonstrate that nTCR successfully recovers the ground-truth reductions even in this more general setting: the consistency loss approaches zero and the identification losses for both $\tau$ and $\omega$ converge to zero. This validates that our theoretical guarantees (Propositions 4.1 and 4.2) and learning algorithm handle arbitrary nonlinear $h_0$ functions, not just the simplified case. Thank you for this suggestion, which has helped strengthen our paper.
>
>
> ## Linear TCR
>
> Our work makes two contributions: (1) we extend linear TCR to the nonlinear case, proving that certain classes of nonlinear models have unique exact solutions (zero consistency loss), and (2) we demonstrate how nTCR can learn interpretable explanations of RL policies.
> Linear TCR was previously applied only to scientific simulations. The similarities between simulating physical systems and RL environments motivated us to study and adapt this framework to the context of policy explanation. While these points were stated in the contribution paragraph at the end of the introduction, we have now added this clarification in the related work section.
>
> We provide comparisons with linear TCR in Appendices F.2 and F.3. These experiments show fundamental limitations: while linear TCR can identify when variables matter (temporal importance) and their average directional effect, it cannot capture non-monotonic relationships or distinguish between different behavioral modes. For instance, in the Pendulum task, linear TCR misses the critical clockwise vs. counterclockwise asymmetry.
>
> ## Definition of $\eta_{norm}$
>
> We apologize for the oversight. The parameter $\eta_{norm}$ is the regularization weight that controls the strength of the normality regularization in our total optimization objective:
>
> $$L_{total} = L_{cons} + \eta_{norm}L_{norm}$$
>
> where $L_{cons}$ is the consistency loss (Eq. 2.5) and $L_{norm}$ is the normality regularization term (Eq. 4.1). This hyperparameter balances the trade-off between achieving interventional consistency and maintaining Gaussian-like distributions for the high-level cause.
>
> We have added this definition immediately after introducing the total optimization objective in Section 4.1 to ensure clarity for readers. Thank you for catching this omission.
>
> ## Prop. 4.1 and 4.2
>
> These propositions serve two purposes:
>
> 1. **Well-posedness**: They demonstrate that the nonlinear TCR learning problem is not ill-posed. Without such guarantees, we might be attempting to optimize an objective with no good solutions. The propositions show that, at least for certain model classes, perfect interventional consistency (zero loss) is achievable.
>
> 2. **Validation benchmark**: Proposition 4.2 provides analytical solutions (Eq. 4.3) that serve as ground truth. We use these in Section 5.1 to validate that our learning algorithm correctly recovers known solutions (Fig. 2) and to understand the interplay between consistency and normality regularization (Appendix F.1).
>
> **On the assumptions**: As noted in Section 4.2 (L288), we don't expect real RL environments to satisfy these specific conditions (Gaussian noise, particular functional forms). The propositions characterize one class of models for which the existence/uniqueness of exact transformations is guaranteed; other classes likely admit exact reductions too. In practice, optimization of the consistency loss seeks the best approximately consistent reduction, and our RL experiments demonstrate that even imperfect consistency  yields meaningful policy explanations. While exact solutions may be rare, the existence results justify our approach of minimizing consistency loss to find interpretable causal explanations.

---

> > ### Author Response · Authors · 2025-11-22
> > **Response to Reviewer heb9 (2/3)**
> >
> > ## The Choice of Function Class in Section 4.3
> >
> > Our Gaussian kernel basis functions are a standard choice in kernel methods, though we are not aware of their specific use in interpretability work. We chose this representation because it allows us to capture nonlinear relationships while maintaining interpretability: the learned weights directly show which values, at which features and times, most influence policy performance.
> >
> > **Capturing linear patterns**: Yes, our function class can represent linear relationships. If a feature contributes linearly to the high-level cause Z at time t, this would appear as a linearly increasing color gradient along the y-axis in our heatmaps (Figs. 3 and 4). The $\omega$-map for Torque in Policy B (Fig. 3) shows approximately linear behavior around torque=0 for many time steps.
> >
> > **Parameter settings**: The function parameters represent a standard bias-variance trade-off. Wider kernels and fewer kernels both increase smoothing (higher bias, lower variance). Specifically, we set kernel centers $\{\mu_{j,t,k}\}$ equally spaced over $[x^j_{\min}, x^j_{\max}]$ with widths $\sigma_{j,t} = c \cdot (x^j_{\max} - x^j_{\min})/K$, where $K$ is the number of kernels and $c$ is the smoothing parameter.
> >
> > We will add a clarification to Section 4.3 to better explain our design rationale.
> >
> > ## The Necessity of Reduction for Simple Environments
> >
> > While the Pendulum environment has only 3 state variables and 1 action variable, the temporal nature of RL creates a much higher-dimensional explanation problem. With 200 time steps, we are actually analyzing an 800-dimensional low-level causal model to understand which behavioral patterns lead to success or failure.
> >
> > **Validation through simplicity**: We deliberately chose the Pendulum as one of our test cases precisely because its relative simplicity allows us to validate our method's explanations. In such environments, we can cross-check our findings through independent analyses (as shown at the bottom of Fig. 3) and visual inspection of episode videos. This validation is crucial for establishing trust in our method before applying it to more complex scenarios. The fact that our method reveals non-obvious patterns even in this well-understood environment (such as Policy A's unexpected rotational bias despite the environment's symmetry) demonstrates its effectiveness.
> >
> > For higher-dimensional problems, we provided the table tennis experiment with 30+ state variables, showing that our approach scales to more complex robotic control tasks while maintaining interpretability.
> >
> > **Main limitations**: The primary limitation of our approach is the requirement to actively intervene during policy execution. nTCR requires re-running the policy under controlled perturbations to establish causal relationships. This makes our method best suited for scenarios where policies are trained in simulation before real-world deployment; a common practice in robotics and safety-critical applications where understanding failure modes is essential. Our approach can, however, also be integrated in strategies that combine real-world and simulated data [Kang et al., 2019; Rao et al., 2020].

---

> > > ### Author Response · Authors · 2025-11-22
> > > **Response to Reviewer heb9 (3/3)**
> > >
> > > ## The Purpose of Evaluations in Sections 5.2-5.3
> > >
> > > The evaluations in Sec. 5.2-5.3 aim to validate that nTCR produces meaningful explanations of policy behavior. Since "interpretability" is inherently difficult to quantify, we validate our findings by comparing nTCR's outputs against independent analyses of the same policies.
> > >
> > > - **Pendulum (Policy A)**: nTCR identifies a rotational bias. We independently verify this by computing mean rewards for different initial conditions, confirming that pendulums starting in the right quadrant (tending clockwise) achieve significantly higher rewards than those starting left (Fig. 3, bottom middle).
> > >
> > > - **Pendulum (Policy B)**: nTCR suggests that more negative torque would improve performance. We validate this by analyzing trajectories, which show the policy allows over-rotation in the positive $\theta$ direction before applying corrective action (Fig. 3, bottom right).
> > >
> > > - **Table Tennis**: nTCR identifies spatial patterns in policy failure. We cross-check by plotting missed ball positions relative to the racket (Fig. 4c), confirming that balls toward the table edge and at certain heights are indeed more challenging.
> > >
> > > **On accuracy metrics**: While nTCR does achieve lower consistency loss than linear TCR (see Fig. 2 for synthetic experiments), this is expected given its higher function class capacity. It does not show that nTCR is more useful, but that it makes a better low-dimensional approximation of the underlying system. For us, consistency is a means to an end: our primary goal is generating interpretable and actionable insights about policy behavior, which we demonstrate through the independent validations above. For completeness, we have added a plot comparing the consistency losses between linear and nonlinear TCR for the RL experiments in Figure H of the updated manuscript.
> > >
> > > We thank the reviewer again for their review and helping us improve our manuscript.
> > >
> > >
> > > **References**:
> > >
> > > [Kang et al., 2019] Kang, Katie, et al. "Generalization through simulation: Integrating simulated and real data into deep reinforcement learning for vision-based autonomous flight." 2019 international conference on robotics and automation (ICRA). IEEE, 2019.
> > >
> > > [Rao et al., 2020] Rao, Kanishka, et al. "Rl-cyclegan: Reinforcement learning aware simulation-to-real." Proceedings of the IEEE/CVF Conference on Computer Vision and Pattern Recognition. 2020.

---

### Official Review · Reviewer_pKWp · 2025-10-31

**Soundness:** 3
**Presentation:** 3
**Contribution:** 3
**Rating:** 6
**Confidence:** 3

**Summary:**

This paper studies the explainability of RL policies at the policy level.
It proposes a nonlinear Causal Model Reduction framework that abstracts low-level environment dynamics into a high-level causal model, guided by intervention consistency.
Experiments on synthetic causal systems demonstrate the utility of the approach, and results on a complex control task validate its practical application.

**Strengths:**

The problem addressed is highly important for RL. Framing it from the perspective of policy-level decision-making explainability is both interesting and technically challenging, as it requires linking internal policy mechanisms to long-term performance. Advancing this line of work could meaningfully improve the interpretability, debugging, and reliability of RL systems in complex settings.

The paper is well written and easy to follow, and the theoretical part appears strong.

The core idea is clear: use an intervention-consistency high-level model to represent how the long-term returns are influenced.

**Weaknesses:**

The application domain of the method is not fully clear. Please articulate the types of scenarios where it is intended to be used and detail the constraints introduced by the surjectivity assumption. As a concrete edge case: if the long-term (episodic) return is binary—i.e., a single scalar only at the final step—does the method still apply, and under what conditions could the method be applied?


Additionally, for readability, it would be helpful to include a worked example in the introduction to demonstrate how the learned high-level model explains the impact of a specific action on long-term returns.

**Questions:**

Can you explain Eq. 2.4 more? What do you mean by Eq. 2.4? If I understand correctly, \pi_0, \pi_1 are subsets of the actual state variables. Is it correct?

Do the learned state variables retain their significance across time—i.e., if a variable is important at step t, is it still important at t+1 or t+n?  In stationary MDPs, a common assumption is: the state dimension should be time-invariant [1], and the long-term return is additive[2]. Additionally, there is also a discussion about what kind of state variables could be significant for policy learning [3]. How do these conclusions influence the high-level abstracted causal model learning?

[1] Huang, B., Lu, C., Leqi, L., Hernández-Lobato, J. M., Glymour, C., Schölkopf, B., & Zhang, K. (2022, June). Action-sufficient state representation learning for control with structural constraints. In International Conference on Machine Learning (pp. 9260-9279). PMLR.

[2] Zhang, Y., Du, Y., Huang, B., Wang, Z., Wang, J., Fang, M., & Pechenizkiy, M. (2023). Interpretable reward redistribution in reinforcement learning: A causal approach. Advances in Neural Information Processing Systems, 36, 20208-20229.

[3] Liu, Y., Huang, B., Zhu, Z., Tian, H., Gong, M., Yu, Y., & Zhang, K. (2023). Learning world models with identifiable factorization. Advances in Neural Information Processing Systems, 36, 31831-31864.

---

> ### Author Response · Authors · 2025-11-22
> **Response to Reviewer pKWp (1/2)**
>
> We are grateful for the reviewer's insightful comments and suggestions. We appreciate your positive assessment of our theoretical contributions and the importance of the problem we address. Below, we respond to your main concerns.
>
> ## On the Application Domain and Method Constraints
>
> Application Domain: Our method is designed for scenarios where policies can be safely probed through interventions, primarily in simulators before real-world deployment. While we focus on simulation due to safety and cost, the framework could, in principle, be applied to real systems where controlled interventions are feasible, and frameworks that combine simulated and real data [Kang et al., 2019; Rao et al., 2020]. We have added this comment to the discussion on limitations.
>
> Terminal Rewards: When rewards appear only at the final timestep, our method works perfectly well. We process the entire trajectory of state and action variables jointly, capturing any behavioral patterns throughout the episode that have downstream effects on the terminal reward. Our approach considers the full temporal sequence: as long as actions or states during the episode influence the final outcome, our method will identify these in the reduction functions.
>
> Surjectivity: For our nTCR approach, we do not require the $\tau$ and $\omega$ maps to be surjective. Surjectivity is only a requirement in Definition 2.2 for exact transformations in the theoretical background. Our practical method minimizes approximate interventional consistency (Eq. 2.5) and can learn whatever mappings best capture the causal relationships, regardless of whether they are surjective. In practice, the KL divergence favors an overlap between the mapped low level and the high-level distribution, arguably favoring surjectivity. This makes our approach more flexible than the theoretical exact transformation framework.
>
> ## Technical Clarifications
>
> ### Equation 2.4
>
> You are correct that $\pi(0)$ and $\pi(1)$ denote subsets of the low-level variables. Specifically:
>
> Equation 2.4 defines the constructive transformation maps:
> - $\tau(\mathbf{x}) = (\bar{\tau}\_0(\mathbf{x}\_{\pi(0)}), \bar{\tau}\_1(\mathbf{x}\_{\pi(1)}))$ means the transformation $\tau$ has two components:
>   - $\tau_0$ extracts the target $Y$ using only variables indexed by $\pi(0)$
>   - $\tau_1$ learns the high-level cause $Z$ using only variables indexed by $\pi(1)$
> - $\omega(\mathbf{i}) = (0, \bar{\omega}\_1(\mathbf{i}\_{\pi(1)}))$ means interventions are mapped such that:
>   - The first component is always 0 (preventing direct interventions on the target $Y$)
>   - Only interventions on variables in $\pi(1)$ are mapped to high-level interventions
>
> In our RL application (Sec. 3):
> - $\pi(0)$: indices of all reward variables $\{R_1, ..., R_T\}$
> - $\pi(1)$: indices of all state and action variables $\{S_0, A_0, ..., S_T, A_{T-1}\}$
>
> This partition ensures that:
> 1. The target $Y$ (cumulative reward) is computed only from rewards
> 2. The learned cause $Z$ depends only on states/actions that can influence rewards
> 3. Changes in $Y$ must be explained through the causal pathway $Z \rightarrow Y$, not through direct intervention
>
> ### Temporal Significance of Learned Variables
>
> The interpretable function class in Section 4.3 learns a separate weight for every feature $j$ at time $t$. That is, we resolve the features across time, such that we can tell that a certain value of that feature at a certain time contributes significantly to the success or failure of a policy. See, for example, Fig. 4: we can see that there are two regions of the joint position 0 of the arm that contribute positively (Region A) or negatively (Region B) to the cumulative reward. These regions correspond to good and bad timing of the swinging motion of the arm relative to the time when the ball typically arrives in the hitting distance of the arm.
>
> However, resolving these weights across time is just a modelling choice of the function class and can be changed flexibly within our framework. You could make the feature-value importances invariant across time by encoding this in the reduction functions (Eq. 4.4). We actually experimented with this at some point, but did not find that it gave much benefit for interpretability. We're happy to add a comment about this in the paper.

---

> > ### Author Response · Authors · 2025-11-22
> > **Response to Reviewer pKWp (2/2)**
> >
> > ### Relationship to Existing State Representation Learning literature
> >
> > Thank you for pointing out these papers providing a causal perspective on policy optimisation. In contrast to this work, our analysis does not draw general conclusions about the importance of specific state variables for policy learning: we learn policy-level explanations with a single - already trained - policy, and the causal high-level variables that we extract are related to this specific policy. That means that the variables relevant to policy learning have already been selected during training. As illustrated in Figure 3, we can observe that two different policies may lead to very different mappings between low- and high-level variables and put emphasis on different time intervals of the experiments. This is because we explain how a specific policy fails or succeeds and the location (in time and in the state-space) of the events leading to those are a priori policy dependent. We added the references to related work to make this clarification as follows.
> >
> > “
> > Another related field is the study of Markov Decision Processes (MDP), which model the systems to be controlled by RL-agents. Several works [ 36– 38 ] have taken a causal perspective to study MDPs and related models in order to analyze policy optimization, and notably the relevance of specific state variables for those policies. These methods can be leveraged to learn causal representations that are suitable for training RL-agents, which contrasts with our goal of interpreting already trained policies.
> > ”
> >
> > We believe these clarifications strengthen our contribution and demonstrate the practical applicability of our method.
> >
> > **References**:
> >
> > [Kang et al., 2019] Kang, Katie, et al. "Generalization through simulation: Integrating simulated and real data into deep reinforcement learning for vision-based autonomous flight." 2019 international conference on robotics and automation (ICRA). IEEE, 2019.
> >
> > [Rao et al., 2020] Rao, Kanishka, et al. "Rl-cyclegan: Reinforcement learning aware simulation-to-real." Proceedings of the IEEE/CVF Conference on Computer Vision and Pattern Recognition. 2020.

---

### Official Review · Reviewer_EFs4 · 2025-11-05

**Soundness:** 3
**Presentation:** 3
**Contribution:** 3
**Rating:** 6
**Confidence:** 2

**Summary:**

This paper presents Nonlinear Targeted Causal Reduction (nTCR), a framework for generating policy-level explanations (PLEs) for reinforcement learning (RL) agents through the lens of causal model reduction (CMR). Building on prior work in Targeted Causal Reduction (TCR) [Kekić et al., UAI 2024], the authors generalize it to nonlinear settings to better capture complex relationships in RL environments.
The core idea is to treat the RL system as a low-level structural causal model (SCM) over states, actions, and rewards, introduce random perturbations to actions as interventions, and learn a high-level, reduced causal model that remains approximately interventional-consistent with the original one.

The paper proves conditions for existence and uniqueness of exact nonlinear causal reductions under additive noise SCMs and introduces normality regularization via Wasserstein distance to maintain interpretability and stability. Experiments on synthetic data, the classic Pendulum control task, and a robotic table tennis simulation show that nTCR can uncover interpretable behavioral patterns and failure modes in RL policies—e.g., biases toward specific trajectories or torque directions.

**Strengths:**

1. **Bridging causality and RL interpretability**:
It connects causal abstraction theory with reinforcement learning policy analysis—a growing but underexplored intersection—and provides a unified, formal framework to derive explanations that are causally meaningful rather than purely correlational.

2. **Conceptually grounded contribution**:
The paper builds upon rigorous causal inference foundations (structural causal models, causal abstraction, targeted reductions) and extends them into nonlinear domains relevant for RL explainability.

3. **Theoretical depth**:
The authors derive uniqueness and existence guarantees for exact transformations under nonlinear SCMs—addressing identifiability, a critical issue in causal representation learning.

4. **Experiments**:
The combination of synthetic validation (matching theory) and applied tasks (Pendulum, robotic table tennis) is well executed. The qualitative analyses show clear, human-understandable insights into policy behaviors and biases.

5. **Interpretable nonlinear design**:
The use of Gaussian-kernel–based function classes for τ (state/action to high-level cause) and ω (intervention mappings) is interesting and helpful, allowing visualization of how specific features and time steps influence outcomes.

6. **Model-Agnosticism**: The approach treats the RL policy as a black box, which means it can be applied to explain any policy, regardless of its architecture, as long as it can be executed in a simulator.

**Weaknesses:**

1. **Dependence on interventional assumptions**:
The method relies on shift interventions on continuous actions and Gaussianity assumptions at the high level, limiting its applicability to discrete or stochastic policy settings. Future work directions mention this but downplay its importance.

2a. **Interpretability trade-off**:
Although Gaussian kernel maps improve interpretability, the framework still produces dense, high-dimensional weight maps that may be difficult to interpret without significant post-hoc visualization. Human usability of these explanations is not evaluated.

2b. _Interpretability of ω Maps_: The paper provides excellent interpretation for the learned τ maps (how states/actions map to the high-level cause). However, the interpretation of the ω maps (how low-level interventions map to high-level ones) is less explored. For the Pendulum, it is simple ("more negative torque is better"), but for more complex actions, interpreting the learned ω map could be challenging.

3. **Limited discussion on uncertainty or robustness**:
The causal reductions’ stability under noise or alternative intervention magnitudes is briefly shown in ablations but not rigorously analyzed.

4. **Single High-Level Cause**: The current formulation focuses on explaining a target variable via a single learned high-level cause Z. While powerful, many complex policy failures might result from the interaction of multiple distinct factors. The paper mentions this as a possible extension but does not elaborate on the challenges, such as ensuring the learned causes are genuinely distinct (disentangled).

---

**Minor Typos**
- Ln 126-127: "whih" $\rightarrow$ "which"

**Questions:**

1. The paper focuses on continuous action spaces where "shift interventions" are naturally defined. You briefly mention extending the framework to discrete actions by perturbing the underlying policy parameters. Could you elaborate on the challenges this might pose? Specifically, how would you ensure the resulting interventions are both meaningful for causal discovery and don't simply push the policy into chaotic, out-of-distribution behavior, especially in high-dimensional state spaces like Atari where the input is an image?
---

2. You correctly identify that the high sample complexity of requiring many perturbed episodes makes nTCR primarily suited for simulated environments. Looking forward, what do you see as the most promising path to applying this powerful explanatory framework in real-world, non-simulated settings where extensive intervention is infeasible? For instance, could it be combined with a learned world model, or could a limited set of real-world interventions be used to fine-tune an explanation initially learned in a less-perfect simulator?

---

3. How sensitive are the learned causal reductions to the choice of intervention strength (σ) and the number of perturbed episodes? Appendix Figure A shows how σ was selected based on this trade-off. How robust are the final learned explanations (i.e., the τ and ω heatmaps) to this choice? For example, if you were to select a slightly different σ, would the identified causal patterns remain stable, or are the explanations highly sensitive to this hyperparameter?

---

4. The paper focuses on a single high-level cause. Could you elaborate on the challenges of extending the nTCR framework to discover multiple, distinct high-level causes? Would the normality regularization need to be adapted, for instance, to a multi-modal prior to prevent the causes from collapsing into (a single, less-interpretable composite cause) one?

---

> ### Author Response · Authors · 2025-11-22
> **Response to Reviewer EFs4 (1/2)**
>
> We thank the reviewer for their thoughtful and constructive feedback on our paper. Below, we address your main concerns and questions.
>
> ## Robustness
>
> To address this concern, we have conducted an additional experiment on the Pendulum task where we vary the intervention strength $\sigma$ (the standard deviation of the action perturbations). We find that our main conclusions about policy behavior remain stable across a reasonably wide range of intervention strengths. See Appendix F.5 and Figure I in the updated manuscript.
>
> When interventions are too strong, they push the policy far outside its normal operating regime, leading to episodes that no longer reflect the policy's intended behavior. In this case, the learned reductions become less meaningful, as expected. Conversely, when interventions are too weak, the causal signal becomes difficult to detect above the natural variability in episode outcomes, making it challenging to learn meaningful reduction maps.
>
> Our experiments show that there exists a practical "sweet spot" where interventions are strong enough to generate detectable causal effects but not so strong as to fundamentally alter the policy's behavior. In practice, we select intervention strengths by monitoring their impact on average episode rewards: we choose values that cause modest but measurable performance degradation, ensuring we capture the policy's response to perturbations while remaining within its general operating domain.
>
> ## Scope and Applicability of Our Method
>
> **Discrete action spaces:** We believe there may be a misunderstanding about our suggested extension. We do not propose intervening on neural network parameters, which would indeed lead to chaotic behavior. Instead, policies with discrete action spaces typically output probability distributions over possible actions. Our suggestion for future work is to apply perturbations to these output distributions and observe their effects on cumulative rewards. Concretely, that means the logits of the network implementing the categorical policy can be perturbed by a shift intervention, before being converted to probabilities with the softmax operation. We expect the magnitude of the interventions to be subjected to similar trade-offs as in the continuous case, mentioned in the above “Robustness” section. This would maintain the core principle of our approach (introducing controlled variations to understand causal effects) while adapting to the discrete setting. We clarified the associated sentence in the discussion of the revised manuscript:
> “adaptation to discrete action spaces could be considered, e.g. applying shift interventions to the logits of the network implementing the categorical policy used to sample discrete actions.”
> That said, we believe providing a policy-level interpretability method for continuous actions is a valuable contribution in its own right. Many critical RL applications operate in continuous action spaces, including robotics, autonomous vehicles, and industrial process control. These domains would directly benefit from our current framework.
>
> **Real-world applications:** In principle, our framework can be applied to real-world policies as long as we can intervene on the output actions, which is usually possible. However, our method is primarily aimed at the sim-to-real paradigm commonly used in safety-critical applications, as well as any real-world system where the exploration of the state-space required by RL training from scratch would be too time and resource consuming. The main use is in identifying critical failure modes before policies are deployed in the real world: practitioners can discover dangerous behavioral patterns in the safety of simulation rather than learning about them through costly or hazardous real-world failures. Our approach can however also be integrated in strategies that combine real-world and simulated data [Kang et al., 2019; Rao et al., 2020].
>
> **Computational costs:** In simulation environments, data collection costs are typically manageable. Moreover, the core reduction computation itself has favorable scaling properties. Once episodes are collected, the learning problem reduces to linear regression (Equation 4.4) due to our Gaussian kernel parameterization. This makes the computational cost of learning the reduction maps tractable even for long episodes with many state and action dimensions.

---

> ### Author Response · Authors · 2025-11-22
> **Response to Reviewer EFs4 (2/2)**
>
> ## Interpretability of Learned Reductions
>
> While our reduction maps are indeed high-dimensional, the Gaussian kernel representation provides structure that enables meaningful interpretation. Each weight corresponds to a specific state/action value at a specific time, creating an interpretable "what-when" decomposition of the policy's behavior.
>
> In practice, we observe that the learned weight maps exhibit sparsity due to the applied weight decay: many weights are near zero, highlighting the most influential state-action patterns. This sparsity, combined with our heatmap visualizations, makes behavioral patterns immediately visible. For instance, in Figure 3, the distinct red (positive contribution) and blue (negative contribution) regions clearly reveal Policy A's rotational bias without requiring complex post-processing or additional interpretation tools.
>
> The temporal structure of our representations also aligns with the intuitive understanding of the tasks. In the table tennis experiments (Figure 4), the highest-magnitude weights concentrate around the time of ball contact, exactly when precise control matters most.
>
> We believe these visualizations demonstrate that despite the high dimensionality, our approach successfully distills complex policy behaviors into interpretable patterns that practitioners can understand and act upon. The important part is not the raw number of parameters, but how they are structured and visualized to show meaningful insights about policy behavior.
>
>
> ## Extensions to Multiple High-Level Causes
>
> Having one high-level cause is a deliberate interpretability-expressiveness tradeoff that favors explainability.
> Our method creates a "causal complexity bottleneck" by mapping the complex low-level system to just two variables: a scalar cause $Z$ and the cumulative reward $Y$. This simplification enables clear interpretation, i.e., increases in $Z$ directly correspond to better performance.
> With multiple causes, practitioners would need to reason conditionally about when each cause matters and how they interact, making explanations less actionable.
> While we acknowledge that some complex failures may require multiple causes, we believe our single-cause approach captures the most influential behavioral patterns in a way that practitioners can immediately understand and act upon. Future work could explore partitioning the low-level cause variables beyond $\pi(1)$ into multiple subsets, similar to the linear TCR framework, to identify multiple distinct causes while maintaining interpretability within each partition.
>
> **References:**
>
> [Kang et al., 2019] Kang, Katie, et al. "Generalization through simulation: Integrating simulated and real data into deep reinforcement learning for vision-based autonomous flight." 2019 international conference on robotics and automation (ICRA). IEEE, 2019.
>
> [Rao et al., 2020] Rao, Kanishka, et al. "Rl-cyclegan: Reinforcement learning aware simulation-to-real." Proceedings of the IEEE/CVF Conference on Computer Vision and Pattern Recognition. 2020.

---

### Author Response · Authors · 2025-11-22
**Response to All Reviewers**

We sincerely thank all reviewers for their thoughtful and constructive feedback on our paper.

We were pleased to see that reviewers recognized both the importance and quality of our work. All reviewers acknowledged that we tackle a significant problem, noting "the problem addressed is highly important for RL" (pKWp) and describing it as "well motivated and relevant" (heb9). The reviewers appreciated our causal approach to RL explainability, emphasizing that "framing it from the perspective of policy-level decision-making explainability is both interesting and technically challenging" (pKWp) and that "advancing this line of work could meaningfully improve the interpretability, debugging, and reliability of RL systems in complex settings" (pKWp).

Our theoretical contributions were consistently praised across reviews. Reviewers found "the theoretical contribution seems rigorous" (heb9), while highlighting our "theoretical depth" (EFs4), including "uniqueness and existence guarantees" (EFs4). Multiple reviewers appreciated our framework's clarity, noting "the core idea is clear" (pKWp) and emphasizing that we provide "a unified, formal framework to derive explanations that are causally meaningful rather than purely correlational" (EFs4).

The practical aspects of our work also received positive feedback. Reviewers noted that our experimental scope "is well executed" (EFs4) with "qualitative analyses [that] show clear, human-understandable insights into policy behaviors and biases" (EFs4). They particularly valued our "interpretable nonlinear design" (EFs4) and the "model-agnosticism" (EFs4) of our approach, which "can be applied to explain any policy, regardless of its architecture" (EFs4). The overall presentation was also well-received, with reviewers finding "the paper is well written and easy to follow" (pKWp).

The main concerns raised by reviewers include:

1. **Experiments**: Comparison with linear TCR (heb9), the setting of the synthetic experiments (heb9), and limited analysis of the causal reductions' stability under different intervention magnitudes (EFs4).

2. **Framework assumptions**: Questions about the surjectivity assumptions and their constraints (pKWp), clarification needed on the constructive transformations in Eq. 2.4 (pKWp), dependence on continuous action spaces and shift interventions (EFs4), and the current focus on a single high-level cause (EFs4).

3. **Interpretability considerations**: Questions about the interpretability of the reduction maps and the trade-off between expressivity and interpretability (EFs4, heb9).


In response to these reviews, we have:

1. **Conducted additional robustness experiments**: We added new experiments analyzing the stability of our method under different intervention strengths for the Pendulum task, demonstrating that our approach produces consistent explanations across a practical range of perturbation magnitudes (Appendix F.5).

2. **Extended and strengthened our synthetic experiments**: We expanded our validation experiments to test more general cases of the theoretical framework, including systems with multiple low-level variables making up the target and nontrivial causal relationships among them, confirming identification of the ground truth solution works for more general low-level systems.

3. **Improved clarity throughout the manuscript**: Based on reviewer feedback, we added clarifications on technical definitions, expanded explanations of our design choices and their trade-offs, and clarified the scope of our method's applicability (including potential extensions to discrete action spaces).

We address specific concerns in detail in our individual responses to each reviewer. We sincerely thank all reviewers for their constructive and detailed feedback, which has greatly helped us strengthen our paper. We believe these revisions have significantly improved the clarity, rigor, and completeness of our manuscript.

---

### Author Response · Authors · 2025-12-03
**Statement to Area Chair (1/2)**

Dear Area Chair,

We understand that the recent circumstances have placed you in a challenging position, having to evaluate our paper without the benefit of reviewer-author discussion. To help you assess whether our revisions adequately address the reviewers' concerns, we provide this summary of the changes we have made and clarifications we have offered.

## Summary of Reviews

Our paper received three reviews:
- **Reviewer EFs4**: Rating 6/10 (marginally above acceptance)
- **Reviewer pKWp**: Rating 6/10 (marginally above acceptance)
- **Reviewer heb9**: Rating 4/10 (marginally below acceptance)

All three reviewers acknowledged the importance of the problem and the rigor of our theoretical contributions. The main concerns centered on experimental evaluation, clarity of certain technical aspects, and questions about the method's scope and applicability.

---

## Changes Made to the Paper

### 1. New Robustness Experiments (Appendix F.5, Figure I)

In response to Reviewer EFs4's concern about stability under different intervention magnitudes, we conducted additional experiments on the Pendulum task varying the intervention strength $\sigma$. These experiments demonstrate that our method produces consistent explanations across a practical range of perturbation magnitudes. We show there exists a "sweet spot" where interventions are strong enough to generate detectable causal effects but not so strong as to fundamentally alter policy behavior.

### 2. Extended Synthetic Experiments (Section 5.1, Figure 2)

Reviewer heb9 raised a valid point about our synthetic experiments using $h_0 \equiv 0$. We have updated Section 5.1 to use $|\pi(0)| = 2$ with nontrivial $h_0$ functions sampled from the same neural network class used for other causal mechanisms. The results confirm that nTCR successfully recovers ground-truth reductions even in this more general setting, validating that our theoretical guarantees handle arbitrary nonlinear $h_0$ functions.

### 3. Added Definition of $\eta_{\text{norm}}$

Reviewer heb9 correctly noted that the parameter $\eta_{\text{norm}}$ was not explicitly defined. We have added its definition immediately after introducing the total optimization objective in Section 4.1: $\eta_{\text{norm}}$ is the regularization weight controlling the strength of normality regularization in $\mathcal{L}\_{\text{total}} = \mathcal{L}\_{\text{cons}} + \eta\_{\text{norm}} \cdot \mathcal{L}\_{\text{norm}}$.

### 4. Added Consistency Loss Comparison (Figure H)

To provide quantitative comparison between linear and nonlinear TCR for the RL experiments, we added Figure H showing consistency losses for both methods. This supplements the qualitative comparisons already present in Appendices F.2 and F.3.

### 5. Clarified Discrete Action Space Extension

We clarified the discussion of extending our method to discrete action spaces. Rather than perturbing neural network parameters (which would lead to chaotic behavior), we suggest applying shift interventions to the logits of categorical policies before the softmax operation. The revised text now reads: "adaptation to discrete action spaces could be considered, e.g. applying shift interventions to the logits of the network implementing the categorical policy used to sample discrete actions."

### 6. Added References to Related Work

In response to Reviewer pKWp's question about the relationship to existing state representation learning literature, we added references and clarification distinguishing our policy-specific explanations from methods studying general state variable importance for policy learning.

---

> ### Author Response · Authors · 2025-12-03
> **Statement to Area Chair (2/2)**
>
> ---
>
> ## Clarifications of Misunderstandings
>
> ### Linear TCR Comparisons Already Present
>
> Reviewer heb9 stated there was "no sufficient evaluation comparing previous approaches" and requested comparison between TCR and nTCR. However, these comparisons were already present in our submission in **Appendices F.2 and F.3**. These sections show that:
>
> - Linear TCR can identify temporal importance and average directional effects
> - Linear TCR cannot capture non-monotonic relationships or distinguish between different behavioral modes
> - For Policy A in Pendulum, linear TCR misses the critical clockwise vs. counterclockwise asymmetry that nTCR reveals
>
> We believe the reviewer may have overlooked these appendix sections.
>
> ### Surjectivity Assumption
>
> Reviewer pKWp asked about constraints from the surjectivity assumption. We clarified that surjectivity is only required in Definition 2.2 for *exact* transformations in the theoretical background. Our practical nTCR method minimizes approximate interventional consistency (Eq. 2.5) and can learn whatever mappings best capture causal relationships, regardless of surjectivity.
>
> ### Purpose of RL Evaluations
>
> Reviewer heb9 found the purpose of Sections 5.2-5.3 unclear, expecting accuracy comparisons. We clarified that since "interpretability" is inherently difficult to quantify, we validate our findings by comparing nTCR's outputs against independent analyses:
>
> - **Pendulum Policy A**: nTCR identifies rotational bias → independently verified by computing mean rewards for different initial conditions (Fig. 3)
> - **Pendulum Policy B**: nTCR suggests more negative torque would help → validated by trajectory analysis showing over-rotation in positive $\theta$ direction
> - **Table Tennis**: nTCR identifies spatial patterns in failure → cross-checked by plotting missed ball positions (Fig. 4c)
>
> In addition, we also added the accuracy comparison in terms of achieved consistency loss in Fig. H.
>
> ---
>
> ## Summary
>
> We believe our revisions address all substantive concerns raised by the reviewers:
>
> | Concern | Response |
> |---------|----------|
> | Robustness to intervention strength | New experiments in Appendix F.5 |
> | Limited synthetic experiment scope | Extended experiments to general case with nontrivial $h_0$ |
> | Missing $\eta_{\text{norm}}$ definition | Added to Section 4.1 |
> | No comparison with linear TCR | Already in Appendices F.2/F.3; added Figure H |
> | Unclear application domain | Clarified simulation-based focus and sim-to-real paradigm |
> | Surjectivity constraints | Clarified this applies only to exact transformations |
> | Interpretation of Eq. 2.4 | Provided detailed explanation of partition structure |
> | Single high-level cause limitation | Explained as deliberate interpretability-expressiveness tradeoff |
>
> The reviewers consistently praised our theoretical contributions, the importance of the problem, and the quality of our experimental demonstrations. With the clarifications and additional experiments provided, we believe our paper makes a solid contribution to the intersection of causal inference and reinforcement learning explainability.
>
> We thank you for your time in evaluating our submission under these unusual circumstances.
>
> Sincerely,
> The Authors

---

### Meta-Review · Area_Chair_PsRy · 2026-01-07

**Summary:**

The paper proposes Nonlinear Targeted Causal Reduction (nTCR), a framework designed to explain reinforcement learning (RL) policies by abstracting complex low-level dynamics into simplified high-level causal models. The reviewers generally agreed on the high importance and motivation of the problem, noting that bridging causal abstraction with RL interpretability is a significant and technically challenging contribution. The primary concerns that informed the initial ratings included: (1) a lack of quantitative comparison between the proposed nonlinear method and existing linear TCR; (2) concerns that the initial synthetic experiments were too simplistic to validate the theoretical guarantees; (3) questions regarding the stability of the learned explanations under varying intervention magnitudes; (3) technical ambiguities regarding the surjectivity assumption, specific parameter definitions, and the interpretation of high-level mappings.

**Reviewer Concerns:**

The authors provided a comprehensive rebuttal and manuscript revision that addressed nearly all technical and evaluative concerns:

### **Addressed Concerns:**
*
**Evaluation Gap:** The authors added Figure H, providing a quantitative consistency loss comparison between linear and nonlinear TCR, and highlighted existing qualitative comparisons in the appendices showing where linear models fail to capture behavioral modes (e.g., rotational bias in Pendulum) .

**Synthetic Validation:** Section 5.1 was updated to include non-trivial  functions, confirming that nTCR recovers ground-truth reductions in more general settings .

**Robustness to Interventions:** New experiments in Appendix F.5 demonstrate that learned explanations remain stable across a practical sweet spot of intervention strengths .
*
**Technical Definitions:** The authors explicitly defined  and clarified that surjectivity is a theoretical requirement for exact transformations rather than a constraint on the practical nTCR optimization .
*
**Discrete Action Spaces:** A clear path for extending the framework via logit-shift interventions was provided, addressing the applicability to categorical policies .


### **Outstanding Concerns:**

**Human Usability:** Reviewer EFs4 noted that while the Gaussian kernel maps provide structure, the high-dimensional weight maps still require significant visualization effort, and their utility for human users has not been formally evaluated .

**Single Cause Limitation:** The current focus on a single high-level cause () is a deliberate trade-off for interpretability, but as noted by reviewers, it may not capture the full complexity of certain multi-factor policy failures .

**Reviewer Scores:**

Based on the depth of the rebuttal and the addition of requested experiments, the scores would likely have shifted as follows:

* **Reviewer EFs4 (Initial 6/10):** Would likely maintain a **6/10 or move to a 7/10**. They praised the "theoretical depth" and "interpretable nonlinear design". The authors' detailed response on robustness and discrete actions directly addressed their primary "Questions" and "Weaknesses".


* **Reviewer pKWp (Initial 6/10):** Would likely move to a **7/10**. Most of this reviewer's concerns were related to "Application Domain" and "Technical Clarifications" (e.g., Eq 2.4 and surjectivity) . The authors' exhaustive explanation of the variable partitions and the sim-to-real paradigm resolved these points .


* **Reviewer heb9 (Initial 4/10):** Would likely move to a **6/10**. This reviewer's "main concern" was the lack of comparison to previous approaches and the simplicity of synthetic data . By adding Figure H (quantitative comparison) and Figure 2 (extended synthetic results), the authors neutralized the core reasons for the initial rejection.


The authors have successfully demonstrated that nTCR provides a theoretically grounded and practically useful framework for RL explainability. By extending causal model reduction to nonlinear settings and proving uniqueness/existence guarantees, the paper makes a solid theoretical contribution. The added experimental comparisons and robustness tests effectively address the reviewers' skepticism regarding the method's superiority over linear baselines and its stability in practice. While the "single-cause" limitation and "human-usability" evaluation remain as areas for future work, the current submission is a rigorous and well-motivated advancement in Causal Representation Learning for RL.

---

### Decision · Program_Chairs · 2026-01-26

Accept (Poster)